# Effects of Different Farming Models on Muscle Quality, Intestinal Microbiota Diversity, and Liver Metabolism of Rice Field Eel (*Monopterus albus*)

**DOI:** 10.3390/foods14132383

**Published:** 2025-07-05

**Authors:** Yifan Zhao, Wenzong Zhou, Muyan Li, Yuning Zhang, Weiwei Lv, Weiwei Huang, Hang Yang, Quan Yuan, Mingyou Li

**Affiliations:** 1Key Laboratory of Exploration and Utilization of Aquatic Genetic Resources by the Ministry of Education, Shanghai Ocean University, Shanghai 201306, China; 17854232836@163.com (Y.Z.); limuyan0720@163.com (M.L.); 2Eco-Environmental Protection Research Institute, Shanghai Academy of Agricultural Sciences, Shanghai 201403, China; zhouwz001@163.com (W.Z.); ynzhang@saas.sh.cn (Y.Z.); wwlv@saas.sh.cn (W.L.); hwwswx@163.com (W.H.); yhangqu2024@163.com (H.Y.);; 3Key Laboratory of Integrated Rice-Fish Farming, Ministry of Agriculture and Rural Affairs, Shanghai Academy of Agricultural Sciences, Shanghai 201403, China; 4Wuxi Fisheries College, Nanjing Agricultural University, Wuxi 214081, China

**Keywords:** *Monopterus albus*, farming mode, muscle nutrition, intestinal microbiota, liver metabolomics

## Abstract

As consumer demand for quality fish products continues to rise, quality has become a key factor in market competition. Ecological aquaculture research is exploring various farming methods to balance high-quality demand with environmental protection. This study compared three aquaculture models—cage culture (CG), recirculating aquaculture (RAG), and rice–fish co-culture (RG)—by analyzing muscle quality (AOAC, GC-MS), intestinal microbiota (16S rRNA), and liver metabolism (LC-MS) to assess their effects on *M. albus*. In terms of muscle quality, the RG group showed increased levels of EPA and DHA, reduced muscle moisture and crude lipid content, and enhanced crude protein accumulation. The crude protein content was significantly higher in the RAG group than in the CG group (*p* < 0.05). The RG group also had the highest levels of total, essential, and umami amino acids, followed by the RAG and CG groups. In terms of intestinal microbiota, the RG group had the highest microbial diversity and stability, with increased abundance of Firmicutes and Bacteroidetes and decreased levels of Proteobacteria. Compared to the CG, the RAG group also showed increased microbial diversity and a reduction in pathogenic genera. Liver metabolomics analysis demonstrated that the RG group had significant advantages over the CG group in amino acid, lipid, and energy metabolism. The RAG group exhibited upregulation of glycerophospholipid metabolism and a decrease in oxidative stress marker levels. Overall, the RG group enhanced muscle quality and optimized intestinal and liver metabolism in *M. albus*.

## 1. Introduction

With the rapid growth of the global population and increasing demand for high-quality protein, aquatic products have become an important means of meeting human food needs [1]. Fish are an important part of the human diet and are a key source of high-quality animal protein. Research has shown that the delicious taste and rich nutritional value of fish positively influence consumers’ purchasing decisions [2]. At the same time, consumers are increasingly concerned not only with taste and nutrition, but also with whether the farming process is environmentally friendly and aligned with sustainable development principles [3]. Therefore, effective aquaculture models that deliver superior meat quality not only meet these demands but also generate significant economic benefits and market competitiveness, earning strong consumer approval. However, the rapid expansion of global aquaculture has led to issues such as environmental pollution and resource waste becoming increasingly evident. Traditional intensive aquaculture models have increased aquatic production. However, they have also engendered numerous environmental and health concerns, including water quality degradation [4], frequent disease outbreaks [5], antibiotic abuse, and food safety risks [6]. This not only threatens the stability of aquatic ecosystems and affects the sustainable development of aquaculture, but also impacts the nutritional value and freshness of fish meat, making it difficult for consumers to accept. To address these challenges, there is an urgent need to explore greener and more sustainable farming models. In recent years, recirculating aquaculture systems (RAS) and rice paddy aquaculture have emerged as key research hotspots in modern aquaculture because of their advantages in enhancing resource efficiency and reducing environmental pollution [7,8].

Although the cage culture system can increase the stocking density per unit water volume, limited water circulation often leads to water quality deterioration, especially under high-density conditions, which increases the risk of disease and ecological imbalance. In contrast, RAS maintains stable water quality and reduces disease incidence through continuous water circulation and purification processes. These systems use biological and physical filtration to remove waste and harmful substances, thereby significantly reducing water consumption compared with traditional methods [9]. However, despite its environmental advantages, the high initial investment and complex management requirements of RAS remain major barriers to its widespread adoption [10]. In contrast, the rice-fish co-culture system (RFCS), a green ecological model, leverages synergies between rice cultivation and aquaculture to promote self-purification of the farming environment, reduce external inputs, and enhance resource recycling efficiency. By integrating rice planting with aquatic animal husbandry, the RFCS achieves dual utilization of paddy fields and concurrently increases the yields of rice and fish [11]. The paddy environment provides unique growth conditions for aquatic species, characterized by abundant natural food and high-quality water [12]. Meanwhile, the activities of aquatic animals improve the soil structure in rice fields and suppress weeds and pests, fostering the collaborative development of agriculture and aquaculture. The natural feed resources and rich water environment in rice fields create favorable conditions for the growth and development of rice field eels (*Monopterus albus*), contributing to the improvement of their muscle nutritional value and the accumulation of flavor substances, thereby enhancing overall farming efficiency and market acceptance. As such, RFCS is regarded as a sustainable agricultural solution that balances productivity and ecological integrity [13].

The rice field eel (*Monopterus albus*), also known as the Asian swamp eel, is an important traditional aquaculture species in China. Characterized by an elongated, slippery body with distinct yellow-brown stripes, it is nutritionally valued for its high protein and low fat content, making it a key species in freshwater aquaculture [14]. In 2023, the farmed output reached 355,200 tons [15]. Consumer demand for high-quality *M. albus* continues to increase; therefore, exploring eco-friendly and cost-efficient aquaculture models to enhance farming efficiency is critical. Traditional *M. albus* farming has largely relied on cage culture; however, this model faces challenges such as water pollution and resource waste. Traditional cage aquaculture, due to excessive stocking densities, exacerbates the transmission of pathogens, such as bacteria and parasites, compelling farmers to rely heavily on antibiotics and pesticides [6]. Although this approach may enhance economic returns, it simultaneously triggers significant environmental pollution. Moreover, cage structures and flotation devices are predominantly made of plastic materials, which can release microplastic particles and leach harmful substances, such as heavy metals and biocides, into the surrounding water upon prolonged submersion. These factors collectively contribute to ecological degradation [16]. Therefore, the development of environmentally sustainable aquaculture systems has become increasingly imperative. With the promotion of ecological farming concepts, rice-fish co-culture systems and recirculating aquaculture have emerged as promising research directions. Numerous studies in recent years have highlighted the positive impact of RFCS on the growth and quality of various aquatic species. For example, rice-crab and rice-shrimp co-culture systems enhance the quality and nutritional value of aquaculture products, improving muscle composition (amino acids and fatty acids) in Chinese mitten crabs (*Eriocheir sinensis*) and boosting flavor, umami taste, and palatability in crayfish (*Procambarus clarkii*), while promoting gonadal, intestinal, and digestive health and reducing disease incidence [17,18]. RAS is an efficient water quality management technology that minimizes water consumption and pollutant discharge through water recycling and purification. These systems not only reduce reliance on natural water resources but also create a stable aquatic environment, making them a promising solution for sustainable intensification in aquaculture [9].

Despite the numerous advantages of RFCS observed in practice, systematic research specifically focusing on *M. albus* aquaculture modes remains limited. To date, no study has systematically compared the differences in muscle quality, intestinal microbiota diversity, and hepatic metabolic characteristics of *M. albus* under the three major farming models: cage culture (CG), RAS, and RFCS. This study aims to investigate the effects of these three mainstream aquaculture modes on muscle nutrients, intestinal microbiota diversity, and liver metabolism of *M. albus* through direct comparison. By analyzing these physiological indices, the present study seeks to uncover the underlying mechanisms by which different farming environments influence *M. albus* health and quality. We hypothesized that the *M. albus* l co-culture system (RG) would yield superior outcomes in muscle nutrition, intestinal microbiota diversity, and liver metabolic efficiency compared to cage (CG) and recirculating (RAG) systems, due to its ecological advantages in natural food provision and low-stress environment. This study provides a scientific basis for optimizing future aquaculture practices and promoting the sustainable development of *M. albus* farming.

## 2. Materials and Methods

Animal procedures were strictly in compliance with the regulations outlined in the Statute of the Experimental Animal Ethics Committee of the Shanghai Academy of Agricultural Sciences (Approval Number SAASPZ0520016).

### 2.1. Experimental Location

The experiment involved three aquaculture modes: cage culture group (CG), recirculating aquaculture system group (RAG), and rice-fish co-culture system group (RG). The RG and CG experiments were carried out at the Hongyuanze Aquaculture Cooperative in Xiantao City, Hubei Province, China, located at 40.13° N, 117.04° E, characterized by a subtropical monsoon climate. The recirculating aquaculture system experiment was conducted at the *M. albus* farming base of the Zhuanghang Comprehensive Experimental Station, Shanghai Academy of Agricultural Sciences, located at 30.89° N, 121.38° E, characterized by a subtropical monsoon climate.

### 2.2. Experimental Design and Management

#### 2.2.1. Treatment Groups

The RG established in this study was based on three 2-mu (1333.4 m^2^) paddy fields, divided into a rice cultivation area (50%, 666.7 m^2^), an ecological ditch (10%, 133.3 m^2^), and a reinforced ridge (40%, 533.4 m^2^). Rice was transplanted at a density of 15 cm × 20 cm to balance light energy utilization and the connectivity of aquatic habitats. The ditch, with a depth of 0.8 m and a 30 cm clay impermeable layer lining, served as the core habitat for *M. albus*. The ridge was compacted at a 45° slope ratio, with 10 cm diameter PVC pipes embedded every 2 m to facilitate *M. albus* migration between the paddy field and the ditch. A 60-mesh nylon anti-escape net (aperture 0.25 mm) was installed at the water inlets/outlets to prevent the eels from escaping with the water flow. Three cage culture units served as the control group, constructed from 60-mesh polyethylene nets into square open-top fixed cages (1.2 m height, 2 m × 2 m length/width). Bamboo frames were set up in the pond to horizontally fix the cages, which were submerged 80 cm underwater. Each cage was filled with Alternanthera philoxeroides (water peanut), and a distance was maintained between the cages for easy feeding and daily management. The recirculating aquaculture system experiment was conducted at the *M. albus* farming base of the Zhuanghang Comprehensive Experimental Station of the Shanghai Academy of Agricultural Sciences. Three cement tanks (2 m × 2 m, 80 cm water depth) with independent freshwater recirculation systems and one fish nest per tank served as the experimental units. The freshwater recirculation system included a water pump, ultraviolet lamp, and filter materials (biochemical cotton, ceramic rings, and bacterial houses). The fish nest was composed of six S-shaped plastic plates stacked on a grooved plane (1.0 m × 0.5 m × 0.3 m), with a central hole (15–30 cm diameter) serving as the feeding point for *M. albus*.

#### 2.2.2. Replication and Randomisation

All *M. albus* used in the experiment were of the deep-yellow large-spot variety. Healthy *M. albus* of the same cohort (initial weight: 21.05 ± 1.20 g) were randomly allocated across experimental units to minimize selection bias. Specifically, for the rice-fish co-culture system (RG), 2000 individuals were stocked into each of three independent paddy fields (biological replicates), which were spatially separated by >50 m to ensure environmental independence. For cage culture (CG) and recirculating aquaculture (RAG), 200 individuals were randomly assigned to each of three replicate cages and tanks (n = 3 per system). The experiment was conducted from June to October 2024, with rice transplanted on 10 June and *M. albus* stocked on 20 June after 24 h of acclimatization in the respective systems.

#### 2.2.3. Husbandry and Feeding

During the experimental period, *M. albus* was fed a fixed ratio of commercial feed (43% crude protein, 7% crude fat; Hubei Zhaoliang Biotechnology Co., Ltd., Xiantao, China) once daily at 16:00, at a feeding rate of 5% of their body weight. The water quality parameters were monitored weekly throughout the culture period.

### 2.3. Water Quality Monitoring

In the rice–eel co-culture system (RG), the water temperature fluctuated naturally at 26.8 ± 1.5 °C, with a dissolved oxygen (DO) concentration of 5.2 ± 0.8 mg/L, pH of 7.3 ± 0.3, and ammonia nitrogen (NH_3_–N) concentration of 0.15 ± 0.05 mg/L. The cage culture group (CG) was maintained under similar thermal and chemical conditions (27.1 ± 0.7 °C; DO: 5.3 ± 0.4 mg/L; pH: 7.3 ± 0.1; NH_3_–N: 0.15 ± 0.05 mg/L). In contrast, the recirculating aquaculture system (RAG) provided more stable environmental conditions, maintaining a water temperature of 27.2 ± 0.5 °C, with higher dissolved oxygen (6.8 ± 0.2 mg/L), neutral pH (7.1 ± 0.1), and a significantly lower ammonia nitrogen concentration (<0.05 mg/L).

### 2.4. Sample Collection

On October 20^th^, after 24 h of fasting, five healthy *M. albus* were selected for each biological replicate (RG: three independent paddy fields; CG: three cages; RAG: three tanks; total n = 15 per system). The sex of all individuals was preliminarily identified as male based on external morphology and anatomical examinations. The average body weight was 68.7 ± 2.3 g. All individuals were 10 months of age. The eels were anesthetized with MS-222 (Shanghai Reagent Company, Shanghai, China). The eels were dissected under aseptic conditions, and the foregut, intestinal contents, and liver were excised using sterilized scalpels. A portion of the foregut was placed in Bouin’s solution for histological morphology observation, while the intestinal contents and liver from each eel were stored in cryotubes and preserved in liquid nitrogen for intestinal microbiota and hepatic metabolomics analyses. Subsequently, muscle samples were collected from both sides of the body using sterilized scalpels. One portion of the muscle tissue was fixed in 4% formaldehyde for muscle morphology observation, and the remaining muscle was frozen at −20 °C for analysis of basic muscle nutritional components, fatty acid profiles, and amino acid profiles. For muscle morphology analysis, three tissue samples were collected from each of the fish.

### 2.5. Sample Analysis

#### 2.5.1. Nutritional Content of Muscle

The proximate content of muscle was analyzed according to standard AOAC procedures [19]. Moisture content was determined by drying the samples at 105 ± 5 °C until a constant weight was achieved. The crude ash content was measured by incineration at 550 °C for 6 h. The crude lipid content was extracted using a chloroform–methanol mixture, and the crude protein content was determined using the Kjeldahl method, with the nitrogen content multiplied by 6.25 to calculate the crude protein [20].

For the analysis of amino acid content, 40–50 mg of freeze-dried muscle tissue was hydrolyzed in 6 mol/L hydrochloric acid at 110 °C for 24 h. After hydrolysis, 1 mL of the supernatant was transferred into a vial and analyzed using a Hitachi L-8900 automatic amino acid analyzer (Tokyo, Japan) [21].

The fatty acid content was determined using gas chromatography–mass spectrometry (GC-MS; Agilent 7980B, Santa Clara, CA, USA). Approximately 0.3–0.5 g of muscle tissue was weighed into a 10 mL centrifuge tube and homogenized with 5 mL of a methanol–chloroform mixture (1:2, *v*/*v*). The homogenate was filtered, followed by the addition of 4 mL deionized water and centrifugation at 2300 × g for 5 min at 4 °C. The supernatant was discarded, and the lower phase was concentrated. The lipids were then dissolved in 1 mL of chromatographic-grade n-hexane and reacted with 1 mL of 0.4 mol/L potassium hydroxide–methanol solution for methylation over 30 min. After phase separation by adding 2 mL of deionized water, the upper layer was collected for GC analysis. The fatty acid content was quantified using the area normalization method [22].

#### 2.5.2. Muscle and Intestinal Tissue Morphology

Muscle samples were preserved in 4% formaldehyde for a minimum of 48 h, followed by a graded ethanol dehydration process, clearing with xylene, and embedding in paraffin blocks. Longitudinal sections (5 μm thick) were prepared using a microtome (RM 2235; Leica, Wetzlar, Germany) and stained with hematoxylin and eosin (H&E). The morphological characteristics of the muscle tissue were visualized using a light microscope (YS100; Nikon, Tokyo, Japan). Muscle fiber diameter and density were quantified following the methodology described by Yang et al. [23].

The foregut samples were fixed in Bouin’s solution for at least 48 h, dehydrated with ethanol, made transparent with xylene, embedded in paraffin, sectioned (8 μm), and stained with H&E. The sections were photographed under a YS100 optical microscope to observe the intestinal morphology parameters. The number of goblet cells was determined according to the method described by Shi et al. [24].

#### 2.5.3. 16S rRNA Gene Sequencing for Intestinal Microbiota Analysis

Forty-five qualified intestinal content samples from three treatment groups (CG, RG, and RAG; n = 15 per group) were processed alongside blank controls (three sterile swabs for DNA extraction; nuclease-free water for PCR amplification) to monitor contamination.

According to the instructions of the E.Z.N.A.^®^ soil kit (Omega Bio-tek, Norcross, GA, USA), the DNA concentration and purity were examined using a NanoDrop2000 spectrophotometer (Thermo Fisher Scientific, Waltham, MA, USA), and the quality of the DNA was examined using 1% agarose gel electrophoresis. The integrity of the extracted DNA was determined using primers 338F (5′-ACTCCTACGGGGAGGCAGCAG-3′) and 806R (5′-GGACTACHVGGGTWT CTAAT-3′) for the PCR amplification of the V3–V4 The PCR product was extracted from a 2% agarose gel, purified using a PCR Clean-Up Kit (YuHua, Shanghai, China) according to the manufacturer’s instructions, and quantified using QuantiFluor™-ST (Promega, Madison, WI, USA). DNA sequencing and bioinformatic analyses were performed using Illumina’s NovaSeq 6000 platform (San Diego, CA, USA). Raw sequences were processed in QIIME2 (v2023.9) using the DADA2 plugin for denoising (quality threshold Q > 30), merging, and chimera removal. Taxonomic assignment was performed using the SILVA 138.1 database (99% similarity) by Shanghai Personal Biotechnology Co. Ltd. (Shanghai, China). Detailed information on the intestinal microbiota, such as alpha diversity, beta diversity, composition, and abundance, was analyzed using the online Personal Cloud Platform (www.genescloud.cn).

#### 2.5.4. Liquid Chromatography–Mass Spectrometry (LC–MS) Conditions for Metabolomic Analysis

##### Metabolite Extraction

Tissue samples were accurately weighed into 2 mL centrifuge tubes, followed by the addition of 1000 µL of extraction solution consisting of 75% (methanol:chloroform = 9:1) and 25% H_2_O. Three stainless-steel beads were added to each tube. Samples were homogenized using a tissue grinder at 50 Hz for 60 s, and the process was repeated twice.

After homogenization, the samples were sonicated at room temperature for 30 min and subsequently placed on ice for another 30 min. The mixtures were then centrifuged at 12,000 rpm for 10 min at 4 °C. The resulting supernatants were carefully collected and dried under vacuum. The dried residues were reconstituted in 200 μL of 50% acetonitrile containing 2-chloro-L-phenylalanine as an internal standard. The solutions were filtered through a 0.22 µm membrane and transferred to LC-MS vials for subsequent analysis.

##### Liquid Chromatography–Mass Spectrometry (LC-MS) Analysis

Non-targeted metabolomic analysis of liver samples (n = 5/group) from *M. albus* under different aquaculture modes was performed using ultra-performance liquid chromatography-tandem mass spectrometry (UPLC-MS/MS) at Personalbio Technology Co. (Shanghai, China). An ACQUITY UPLC HSS T3 column (Waters, Milford, MA, USA; 100 Å, 1.8 µm, 2.1 mm × 100 mm) was used with a flow rate of 0.4 mL/min, column temperature of 40 °C, autosampler temperature of 8 °C, and injection volume of 2 μL. The mobile phases for the positive/negative ion modes were as follows: mobile phase A was 0.1% formic acid in water, and mobile phase B was acetonitrile (containing 0.1% formic acid). The elution gradient is presented in Table 1.

A Thermo Orbitrap Exploris 120 mass spectrometer, controlled by Xcalibur software (version 4.7, Thermo Scientific), was used for data-dependent acquisition (DDA) mass spectrometry in both positive and negative ion modes. A heated electrospray ionization (HESI) source was employed with the following parameters: a spray voltage of 3.5 kV (positive mode)/−3.0 kV (negative mode), sheath gas at 40 arb, auxiliary gas at 10 arb, a capillary temperature of 320 °C, and an auxiliary gas temperature of 300 °C. The first-order mass spectrometry (MS1) resolution was set to 60,000, with a scan range of 70–1000 *m*/*z*, an AGC Target Standard, and a maximum ion injection time (Max IT) of 100 ms. The top four most intense ions were selected for secondary fragmentation (MS2), with a dynamic exclusion set to 4 s. The MS2 resolution was 15,000, the higher-energy collisional dissociation (HCD) collision energy was 30%, and AGC Target Standard and Max IT Auto were used. The processed data were evaluated using Pareto-scaled principal component analysis (PCA) and orthogonal partial least squares discriminant analysis (OPLS-DA) to identify the overall differences among the three groups. Metabolites with a variable importance in projection (VIP) > 1.0, fold change (FC) > 2 or (FC) < 0.5, and *p*-value < 0.05 (two-tailed Student’s *t*-test) were considered significantly different and were identified as potential metabolic markers. The Kyoto Encyclopedia of Genes and Genomes (KEGG) database was used to investigate the metabolic and signal transduction pathways associated with significantly differentially abundant metabolites across the treatment groups. DAMs were mapped to the KEGG 2023 pathways using MetaboAnalyst 5.0. This study was conducted by Personalbio Technology Co., Ltd. (Shanghai, China). All formal and quality control (QC) samples were analyzed using the chromatographic and mass spectrometric methods described above. Prior to formal injection, 2–4 QC samples were injected for system equilibration, and one QC sample was injected every 5–10 samples during the injection process.

### 2.6. Statistical Analysis

Data were computed using Microsoft Excel (Microsoft Corp., Redmond, WA, USA), and statistical analysis was performed using SPSS 27.0 software (IBM Corp., Armonk, NY, USA). To assess the assumptions for parametric analysis, the data were tested for homogeneity of variance using Levene’s test. Group differences were evaluated using one-way analysis of variance (ANOVA), followed by Tukey’s post hoc test for pairwise comparisons. Data are presented as mean ± SD (standard deviation), with statistical significance determined at *p* < 0.05. For omics data analysis, appropriate software such as MetaboAnalyst 5.0 will be used to ensure proper handling of the data.

## 3. Results

### 3.1. Effects of Different Farming Modes on Muscle Nutrition of M. albus

#### 3.1.1. Muscle Histology

The muscle morphologies of *M. albus* under CG, RAG, and RG modes are shown in Figure 1. Compared to the CG group, the RG group exhibited a significantly increased number of muscle fibers and higher muscle fiber density (*p* < 0.05). Additionally, the muscle fiber density in the RG group was significantly higher than that in the RAG group (*p* < 0.05), while the number of muscle fibers in the RG group was slightly higher than that in the RAG group, although the difference was not significant. The RAG group showed slightly higher muscle fiber number and density than the CG group, but no significant differences were observed between the two groups (Figure 1A–C; Table 2).

#### 3.1.2. Proximate Content of Muscle

As shown in Table 3, significant differences in moisture content were observed among the three groups: the CG group exhibited the highest moisture content, while the RG group had the lowest, which was significantly lower than that in both the CG and RAG groups (*p* < 0.05). No significant difference in crude protein content was found between the RG and RAG groups, but both were significantly higher than that of the CG group (*p* < 0.05). The crude fat content was highest in the CG group, and the RG group had a significantly lower crude fat content than the CG group (*p* < 0.05). The crude ash content was significantly higher in the CG group than in the other two groups (*p* < 0.05). Overall, the CG group showed higher moisture, crude fat, and crude ash levels, whereas the RG group exhibited a higher crude protein content.

#### 3.1.3. Amino Acid Content in Muscle

Muscle nutritional content is a critical indicator of fish muscle quality. Therefore, we analyzed the amino acid and fatty acid profiles of *M. albus* to assess the effects of different aquaculture modes on muscle quality. The results showed that 17 amino acids were detected in all three groups (Table 4). Specifically, the RG group had significantly higher levels of Histidine, Arginine, and Isoleucine than the RAG group (*p* < 0.05). Compared with those in the CG group, the RG group exhibited significantly higher levels of Aspartate, Glutamate, Glycine, Phenylalanine, Histidine, Threonine, Arginine, Valine, Isoleucine, Leucine, and Lysine (*p* < 0.05). The RAG group also had significantly higher levels of Aspartate, Glutamate, Glycine, Histidine, Valine, Lysine, and Leucine than the CG group (*p* < 0.05). Additionally, the total amino acids (TAA), total essential amino acids (TEAA), total non-essential amino acids (TNEAA), and total umami amino acids (TUAA) in the RG group were all significantly higher than those in the CG and RAG groups (*p* < 0.05), following the order RG > RAG > CG.

#### 3.1.4. Fatty Acid Content in Muscle

We further analyzed the fatty acid content of the *M. albus* muscle. A total of 17 fatty acids were detected (Table 5). Except for C13:0, C18:0, C20:0, C24:1, and C20:5, significant differences in the remaining fatty acid components were observed among the CG, RAG, and RG groups. Specifically, the RG group exhibited significantly higher levels of saturated fatty acids (SFA), monounsaturated fatty acids (MUFA), and n-3 polyunsaturated fatty acids (n-3 PUFA) than the other groups, while the levels of n-6 polyunsaturated fatty acids (n-6 PUFA) and total polyunsaturated fatty acids (PUFAs) were significantly lower (*p* < 0.05). In the RAG group, the SFA content was significantly higher than that in the CG group, whereas the n-6 PUFA content was significantly lower (*p* < 0.05); however, no significant differences were observed in MUFA and n-3 PUFA levels between the RAG and CG groups.

### 3.2. Effects of Different Aquaculture Modes on Intestinal Tissue Morphology of M. albus

To further explore the effects of different aquaculture modes on the intestinal structure and immune function of *M. albus*, HE-stained sections of the anterior intestine were examined and compared among the groups. The intestinal histological findings are summarized in Table 6 and Figure 2.

Significant differences were observed in villus circumference (VC) and goblet cell abundance (GCA, A/root; defined as the number of goblet cells per intestinal segment). The RAG group showed significantly higher VC and GCA values than the CG group (*p* < 0.05). Moreover, the RG group had significantly higher GCA than the RAG and CG groups (*p* < 0.05) and significantly higher VC than the CG group (*p* < 0.05), although the difference was not significant compared to the RAG group.

### 3.3. Effects of Different Aquaculture Modes on Intestinal Microbiota Diversity of M. albus

#### 3.3.1. Assessment of the Intestinal Microbiota via 16S rRNA Gene Sequencing

Rank-abundance curves illustrate the species richness and evenness. In Figure 3A, the curves extend broadly along the x-axis and exhibit relatively flat shapes, indicating both high richness and an even distribution of species within each group. Figure 3B shows that the curves tended to plateau with increasing sequencing depth, suggesting that the sequencing effort was sufficient to capture the majority of species diversity.

#### 3.3.2. Analysis of Microbial Diversity OTUs Across All Samples

To investigate the species composition of each sample, operational taxonomic unit (OTU) clustering and taxonomic annotation were performed, and a Venn diagram was created. As shown in Figure 4, the RG, CG, and RAG groups shared 20 common OTUs, with 2954 unique OTUs in the RG group, 575 unique OTUs in the CG group, and 1726 unique OTUs in the RAG group.

#### 3.3.3. Intestinal Microbiota Diversity Analysis

Alpha diversity analysis was performed to assess the diversity of the microbial community. As shown in Figure 5, there were no significant differences in Good’s coverage among the three groups, with all indices approaching 1, indicating that the bacterial communities had been sufficiently sampled. Compared to the CG group, the RG group exhibited significantly higher Chao1, Observed species, Pielou’s evenness (Pielou-e), Shannon, and Simpson indices (*p* < 0.05). The Shannon index was significantly higher in the RAG group than in the CG group (*p* < 0.05), while the Chao1, Observed species, Pielou-e, and Simpson indices showed no significant differences, although the trends suggested that the RAG mode promoted higher richness and diversity of intestinal microbiota in *M. albus* than the CG mode. Beta diversity, an index that measures species diversity differences between different microbial communities, revealed structural disparities in microbial composition by comparing species profiles across samples. Principal coordinate analysis (PCoA) and non-metric multidimensional scaling (NMDS) were employed to visualize these differences. In the PCoA and NMDS plots, the distance between samples reflects their similarity in species composition and abundance, with smaller distances indicating higher similarity in microbial community structure and species abundance. As shown in Figure 6A,B, the three aquaculture groups exhibited no significant overlap and were well-separated in both PCoA (*p* = 0.025) and NMDS (*p* = 0.014), confirming that these spatial separations reflected statistically meaningful differences in microbial community structure, rather than random variation, demonstrating distinct microbial community structures among the groups.

#### 3.3.4. Microbial Species Analysis

To investigate the effects of different farming modes on the intestinal microbiota of *M. albus*, we analyzed the composition of the intestinal microbiota at the phylum and genus levels in the three groups. The results showed that at the phylum level (Figure 7A and Table 7), the top 10 dominant phyla identified were Firmicutes, Bacteroidota, Proteobacteria, Fusobacteriota, Actinobacteriota, Spirochaetota, Acidobacteriota, Chloroflexota, Desulfobacterota, and Verrucomicrobiota. The dominant phyla in the CG group were Firmicutes (23.49%) and Proteobacteria (27.66%). In the RAG group, the dominant phyla were Firmicutes (22.58%) and Bacteroidota (20.66%), whereas the RG group was characterized by Firmicutes (36.49%) and Bacteroidota (25.50%) as the predominant phyla. Compared with the CG and RAG groups, the RG group exhibited significantly higher abundances of Firmicutes and Bacteroidota and lower Proteobacteria levels (*p* < 0.05). When contrasting RAG with CG, although the level of Proteobacteria was lower and the abundance of Firmicutes and Bacteroidota was higher in RAG, these differences did not reach statistical significance.

At the genus level (Figure 7B and Table 7), the top 10 identified genera were Clostridium_sensu_stricto_1, Cetobacterium, Bacteroides, Romboutsia, Lactobacillus, Plesiomonas, Paludibacter, Aeromonas, Paraclostridium, and Cutibacterium. Compared with the CG and RAG groups, the RG group exhibited higher abundances of Clostridium_sensu_stricto_1, Cetobacterium, Romboutsia, and Paraclostridium, and lower abundances of Bacteroides, Plesiomonas, and Aeromonas. Compared with the CG group, the RAG group displayed higher abundances of Clostridium_sensu_stricto_1 and Romboutsia and lower abundances of Plesiomonas and Aeromonas. The microbial shifts observed in the RG group suggest that the rice-fish co-culture environment enhances intestinal function and nutrient absorption by modulating gut microbiota. This effect likely stems from the unique ecological conditions of the integrated system, which fosters a microbiota conducive to host health.

### 3.4. Analysis of Liver Metabolomics in M. albus Under Different Farming Modes

#### 3.4.1. PCA and OPLS-DA Assessment

To systematically analyze the metabolic regulatory mechanisms of M. albus under different aquaculture modes, we performed a deep comparative analysis of metabolic profiles in liver tissues using untargeted metabolomics technology, starting with PCA, an unsupervised method that reduces data dimensionality to visualize latent structures. This analysis revealed that samples within each group (CG, RG, and RAG) clustered tightly with complete separation and no overlap between groups, indicating significant differences in metabolic characteristics among the groups (Figure 8A–C). To identify the key metabolites driving group separation, OPLS-DA was employed as a supervised method to maximize intergroup differences while minimizing intragroup variations, providing reliable correlation information for differentially abundant metabolites. The model parameters (R^2^ = 0.997, Q^2^ = −0.451 for CG vs. RG; R^2^ = 0.998, Q^2^ = −0.487 for RG vs. RAG; and R^2^ = 0.999, Q^2^ = −0.211 for CG vs. RAG) of R^2^ > 0.5 and Q^2^ < 0 confirmed stable and reliable models, demonstrating significant differences in the metabolic profiles between the groups (Figure 8D–F).

#### 3.4.2. Screening of Differentially Abundant Metabolites

In this experiment, LC-MS was employed to analyze the metabolic components in the livers of three *M. albus* groups under positive and negative ion modes, with differentially abundant metabolites screened based on the OPLS-DA model using the criteria of variable importance in projection (VIP) > 1, fold change (FC) > 2 or FC < 0.5, and adjusted *p*-value (FDR) < 0.05 to control for false discoveries. In the figure, red represents upregulated metabolites, and blue represents downregulated metabolites. The RG group exhibited 4893 upregulated and 206 downregulated metabolites compared to the CG group; the RG group had 2094 upregulated and 159 downregulated metabolites compared to the RAG group; and the RAG group contained 2548 upregulated and 1126 downregulated metabolites compared to the CG group (Figure 9A–C).

#### 3.4.3. Correlation Analysis of Differentially Abundant Metabolites

Correlation analysis of the differentially abundant metabolites was used to examine the consistency in the change trend between metabolites by calculating the Pearson correlation coefficient for all pairwise metabolite comparisons. Metabolite correlations often reveal the synergy of their changes: when the linear relationship between two metabolites strengthens, the correlation coefficient approaches 1 (positive correlation) or −1 (negative correlation). Statistical testing was performed for metabolite associations, with *p* < 0.05 indicating statistical significance. In Figure 10, both the vertical and diagonal axes represent the names of differentially abundant metabolites, where the color denotes the correlation type (red for a positive correlation, blue for a negative correlation), and the color intensity indicates the strength of the correlation (darker colors signify stronger correlations). The pairwise comparisons across the three aquaculture modes are shown in Figure 10A–C.

#### 3.4.4. Metabolic Pathway Analysis of Differentially Abundant Metabolites

To investigate the effects of the three farming modes on the hepatic metabolic pathways of *M. albus*, pairwise comparisons of differentially abundant metabolites were performed, and KEGG pathway enrichment analysis was conducted based on merged positive and negative ion modes (Figure 11A–C). The enrichment significance was assessed using hypergeometric testing with FDR correction, and pathways with adjusted *p* < 0.05 were considered significantly enriched (Appendix A). The results showed that In the comparison between the CG and RG groups, the top 10 metabolic pathways with the highest number of differentially abundant metabolites were biosynthesis of cofactors (12 metabolites), biosynthesis of amino acids (8 metabolites), arachidonic acid metabolism (6 metabolites), carbon metabolism (5 metabolites), glycerophospholipid metabolism (4 metabolites), purine metabolism (4 metabolites), D-amino acid metabolism (4 metabolites), histidine metabolism (4 metabolites), Pantothenate and CoA biosynthesis (4 metabolites), and riboflavin metabolism (3 metabolites) (Figure 11A, Appendix A). In the comparison between the RAG and RG groups, the top 10 enriched pathways included biosynthesis of cofactors (9 metabolites), biosynthesis of amino acids (6 metabolites), arginine and proline metabolism (4 metabolites), glycerophospholipid metabolism (4 metabolites), Pantothenate and CoA biosynthesis (3 metabolites), glutathione metabolism (3 metabolites), D-amino acid metabolism (3 metabolites), riboflavin metabolism (2 metabolites), beta-alanine metabolism (2 metabolites), and glycolysis/gluconeogenesis (2 metabolites) (Figure 11B, Appendix A). For the comparison between the CG and RAG groups, the top 10 pathways were biosynthesis of cofactors (17 metabolites), biosynthesis of amino acids (10 metabolites), carbon metabolism (7 metabolites), sphingolipid metabolism (6 metabolites), glycerophospholipid metabolism (6 metabolites), steroid hormone biosynthesis (6 metabolites), purine metabolism (6 metabolites), riboflavin metabolism (5 metabolites), glycine, serine, and threonine metabolism (5 metabolites), and Pantothenate and CoA biosynthesis (4 metabolites) (Figure 11C, Appendix A). These findings indicate that the biosynthesis of cofactors and amino acids was consistently enriched across all comparisons, highlighting their central roles in metabolic adaptation to different aquaculture environments. Meanwhile, lipid- and energy-related pathways showed mode-specific enrichment, reflecting distinct metabolic reprogramming in the liver of *M. albus* under varying culture conditions (Figure 11A–C and Appendix A).

More than 30 significantly differentially abundant metabolites were screened between the CG and RG groups, with most of them being upregulated in the RG group, and were mainly involved in key metabolic pathways such as biosynthesis of cofactors, biosynthesis of amino acids, arachidonic acid metabolism, carbon metabolism, and glycerophospholipid metabolism. Metabolites like LysoPC(20:4), LysoPC(18:1), LysoPC(20:2), and LysoPC(22:4) showed significant upregulation in glycerophospholipid metabolism (*p* < 0.01), while metabolites related to histidine metabolism, purine metabolism, D-amino acid metabolism, and biosynthesis of amino acids (e.g., Acetylhistamine, Glutathione, and Inosinic acid) tended to be downregulated.

In the comparison between the RAG and RG groups, 20 significantly differentially abundant metabolites were identified, most of which were upregulated in the RG group and were primarily enriched in pathways such as biosynthesis of cofactors, biosynthesis of amino acids, and glycerophospholipid metabolism. Notably, LysoPC(18:1), PC(14:1), and LysoPC 20:5 were significantly upregulated in glycerophospholipid metabolism (*p* < 0.05). Additionally, metabolites in arginine and proline metabolism and beta-Alanine metabolism pathways (e.g., Spermine, Creatinine, and Malonic acid) were upregulated in the RG group, possibly being linked to more frequent movement or muscle tissue activity of *M. albus* in paddy field environments. Conversely, L-Kynurenine, 2,5-diamino-6-(5-phospho-D-ribitylamino)pyrimidin-4(3H)-one, LUMICHROME, and D-(+)-Glucose were significantly downregulated.

In the comparison between the CG and RAG groups, the RAG group showed significant upregulation in Sphingolipid metabolism (e.g., Ceramide, Sphingosine, and Phytosphingosine), hormone intermediates (e.g., DHEA sulfate, and Tetrahydrocorticosterone), and Glycerophospholipid metabolism (*p* < 0.05), while Glutathione and its oxidized products were downregulated. Notably, the biosynthesis of cofactors and amino acid biosynthesis were consistently and significantly enriched across all pairwise comparisons, underscoring their central roles in hepatic metabolic adaptation to different aquaculture environments. Meanwhile, lipid-related pathways, such as glycerophospholipid and sphingolipid metabolism, showed mode-specific enrichment, likely reflecting the structural and functional remodeling of liver membranes in response to environmental conditions.

## 4. Discussion

With the gradual advancement of the aquaculture industry toward a stage of high-quality- development, exploring more ecologically friendly, healthy, and efficient aquaculture modes has become a research hotspot. For *M. albus*, an important economic fish in China, significant quality improvements are required to promote its industrial exploitation. This study compared the effects of three culture modes—cage culture (CG), recirculating aquaculture (RAG), and rice-fish co-culture (RG)—on muscle nutrition, intestinal microbiota, and hepatic metabolism in *M. albus*. The results showed that the RG group outperformed the traditional CG and RAG groups in terms of muscle quality, intestinal microecological stability, and metabolic regulation, indicating that this mode better aligns with the ecological needs and healthy growth of *M. albus*.

Muscle quality is directly linked to the economic value of fish and consumer preferences. Histological analysis revealed that the RG group exhibited a higher number and density of muscle fibers than the CG. Muscle texture characteristics are closely correlated with myofiber density [25]. In fish, myofibers align longitudinally in bundles (myomeres), enveloped by extracellular matrix layers—epimysium, perimysium (PM), and endomysium (EM)—as labeled in Figure 1. As fundamental structural units, myofiber size and quantity vary with species, developmental stage, diet, activity, and environment [26]. Our results indicate that the RG environment optimizes the growth conditions for *M. albus,* promoting tighter myofiber alignment and consequently enhancing muscle quality. Proximate composition analysis showed that the moisture and fat contents in the RG group were lower, while the crude protein content was higher than those in the CG and RAG groups. This indicates that the paddy field environment provides a physiological niche that is more conducive to muscle development, which is consistent with studies on *Lateolabrax japonicus* [27] and *Acanthopagrus schlegelii* [28]. Traditional aquaculture modes, characterized by high dietary feed fat content and limited activity space, often lead to increased fat accumulation. In contrast, the paddy field environment offers *M. albus* a broader activity space, promoting its healthy growth. Similar studies have found that wild fish typically exhibit higher crude protein deposition rates in their muscles because of better food palatability and higher feed conversion efficiency [29]. Research indicates that moisture content plays a critical role in fish muscle quality, with lower moisture levels often associated with higher nutritional value [30].

At the amino acid level, the RG group had higher contents of umami-related amino acids (e.g., glutamic, glycine, and lysine) in muscles than the other two groups, with the highest levels of TAAs, TEAAs, and TNEAAs among the three groups (RG > RAG > CG). Similar studies have shown that rice-fish co-culture systems can significantly improve muscle nutrition and flavor substance accumulation in farmed fish. For example, Wang et al. (2022) reported that tilapia raised in rice-fish co-culture systems had significantly higher amino acid levels than those raised in monoculture ponds [11].

In terms of fatty acid profiles, research indicates that fish flavor and tenderness are influenced by muscle SFA and MUFA content [31]. The RG group exhibited significantly higher SFA and MUFA levels than the other groups. Notably, the n-3 PUFAs (including eicosapentaenoic acid [EPA] and docosahexaenoic acid [DHA]) levels in *M. albus* muscles from the RG group were significantly increased, while n-6 PUFA levels were decreased, thus displaying an optimized fatty acid composition. Evidence shows that excessive intake of n-6 PUFAs may induce abnormal lipid metabolism, pro-inflammatory responses, and oxidative stress, comprising pathophysiological changes recognized as key risk factors for cardiovascular diseases [32]. In contrast, n-3 PUFAs (especially EPA and DHA) play irreplaceable roles in human physiology. They not only effectively regulate inflammatory responses and immune balance but also exert significant cardiovascular protective effects [33].

In contrast, although the RAG system has obvious advantages in water quality control, the RAG group lagged behind the RG group in most muscle indices. This is consistent with the findings in *Acanthopagrus schlegelii* [28], where recirculating aquaculture, despite significantly reducing stress responses, restricted natural food access and behavioral freedom in artificial environments. The limited swimming space in RAG may inhibit muscle development and fatty acid accumulation in *M. albus*.

The intestine is not only one of the most important digestive and absorptive organs in fish, but also comprises the primary immune barrier system. The integrity of its structure and correct function are crucial for maintaining nutrient metabolism and resisting pathogen invasion [34]. Intestinal section analysis revealed significant differences in intestinal morphology and goblet cell distribution among the RG, RAG, and CG groups. The RG group had significantly more goblet cells than the other two groups, and both the RG and RAG groups exhibited larger villus perimeters than the CG group. The villus perimeter is a key indicator of intestinal absorption capacity, with longer villi providing a larger absorptive surface area, while goblet cells secrete digestive fluids to protect the intestine from mechanical damage and microbial invasion [35]. This explains the superior intestinal absorption capacity observed in the RG group. The paddy field environment, by providing a more natural and spacious activity space, facilitates intestinal development and functional optimization in *M. albus*.

As a critical mediator of host-environment interactions, the intestinal microbiota plays a central role in regulating intestinal immune homeostasis, promoting nutrient metabolism, and maintaining host health [36,37]. However, studies have indicated that its community structure, functional diversity, and ecological balance are vulnerable to alterations in aquaculture environments [38]. In terms of intestinal microbiota composition, α and β diversity analyses revealed significant differences in community richness, diversity, and structure among the *M. albus* groups reared under different culture modes. The RG group exhibited significantly higher α-diversity than the CG group, indicating a more complex and stable intestinal microbiota system. The β diversity analysis also showed distinct separation of the microbiota structure among the three groups. Collectively, the α- and β-diversity results highlight the profound impact of culture environments on the intestinal microbiota, suggesting that rice-fish co-culture can enhance the species richness and diversity of the intestinal microbiota of *M. albus*. At the phylum level, the abundance of Firmicutes and Bacteroidota in the RG group was significantly higher than that in the CG and RAG groups, while the abundance of the potentially pathogenic phylum Proteobacteria was significantly lower in the RG group than in the CG group. It has been reported that Firmicutes and Bacteroidota are dominant phyla in healthy intestines, playing critical roles in promoting host health, immunity, and homeostasis [39]. As major components of the intestinal microbiota, Firmicutes (e.g., *Clostridium* and *Lactobacillus*) decompose complex polysaccharides to produce short-chain fatty acids (SCFAs, such as butyric acid), which are essential for maintaining immune homeostasis [40]. In contrast, Bacteroidota excel in degrading dietary fiber, breaking down plant polysaccharides through diverse hydrolytic enzyme systems to generate metabolites like acetic and propionic acids, thereby maintaining microbial balance and inhibiting the colonization of intestinal pathogens [41]. This indicates that the paddy field environment promotes a more beneficial intestinal microbiota structure. Additionally, both the RG and RAG groups showed a significant reduction in the abundance of Proteobacteria compared with that in the CG group. This is consistent with previous findings in which fish with enteritis exhibited decreased Firmicutes and Bacteroidota and increased Proteobacteria abundance, highlighting the link between pathogenic bacterial expansion and impaired gut health [42].

Studies have shown that Clostridium species play important roles in the host metabolic regulatory network by specifically colonizing the intestinal mucosa and continuously producing fermentation end products, such as butyric acid. These processes not only provide energy for intestinal epithelial cells but also help maintain the integrity of the intestinal barrier, prevent pathogen invasion, and promote intestinal immune tolerance [43,44]. Cetobacterium, another key genus for fish health, is critical for glucose metabolism, energy utilization, and immune regulation [45]. Additionally, Plesiomonas is widely present in water bodies and is a common fish pathogen. Its strong environmental adaptability and broad pathogenicity make it a significant threat to fish health in aquaculture settings. Plesiomonas can rapidly reproduce in aquatic environments and exhibit resistance to certain drugs, enabling survival and infection of fish under diverse environmental conditions [46,47]. At the genus level, this study showed that compared with the CG group, the RG and RAG groups had significantly increased abundances of probiotic genera, such as Clostridium_sensu_stricto_1, Paraclostridium, and Cetobacterium, while the abundances of potential pathogenic genera, like Plesiomonas and Aeromonas, were significantly reduced. Moreover, the RG group exhibited significantly higher abundance of these beneficial genera than the RAG group. These changes are closely associated with the microecological complexity of the paddy field system and the input of natural biological foods. In contrast, traditional cage culture (CG) carries a higher disease risk than recirculating aquaculture (RAG) and rice-eel co-culture (RG).

In this study, although the RAG group showed better performance than the CG group for certain bacterial genera, its overall microbiota diversity and structural stability were still inferior to those of the RG group. This suggests that the paddy field system has the advantage of providing a natural microecology and diverse microbial sources. Previous research supports this finding. Wang et al. (2022) observed significantly higher intestinal microbiota diversity in Oreochromis niloticus raised in paddy field systems, in which the rice-fish co-culture group exhibited a significantly higher relative abundance of stress-tolerant bacteria and lower relative abundances of potential pathogens and anaerobic bacteria, helping to reduce pathogenic risks [11]. The paddy field system, with its unique water-soil-plant symbiotic environment, provides a habitat for diverse microbial communities. Interactions among different microbial species in paddy fields form complex food chains and ecological networks, thereby maintaining the health and stability of the ecosystem [48]. Furthermore, the higher microbial diversity and stability observed in the RG group may be closely linked to the environmental complexity of the rice–fish co-culture system. The sediment-rich bottom and aquatic vegetation in paddy fields create a heterogeneous microhabitat, offering diverse environmental microbial sources that promote the establishment of a functionally rich intestinal microbiota [49,50]. Additionally, rice pollen released during the flowering stage provides a natural food source that potentially modulates the gut microbiota composition of *M. albus* [11]. Together, these environmental factors help explain the elevated microbial diversity in the RG group. However, the specific underlying mechanisms require further investigation to be fully understood.

In contrast to the natural ecological environment created by rice-fish co-culture, fish in cage culture systems (CG) experience significantly different environmental pressures. Due to the closed nature of the culture system, monotonous feed, and external stressors, fish are often in a state of persistent stress, which can trigger intestinal microecological imbalance and immune dysfunction [32,51]. In such environments, pathogenic microorganisms gain a competitive advantage, easily occupying core ecological niches and leading to the rapid expansion of pathogenic populations.

Metabolomic analysis using LC-MS technology can systematically decipher changes in metabolic pathways under environmental stress or pathological conditions, effectively reflecting the host’s detailed metabolic responses to different stimuli and environments [52]. Research has shown that, as a key organ for metabolic regulation in fish, liver metabolic processes are closely linked to culture environment, feed composition, and feeding patterns [53,54].

This study employed untargeted metabolomics to compare the hepatic metabolic profiles of the three *M. albus* groups. The results showed that the RG group had 4893 and 2094 upregulated metabolites compared to the CG and RAG groups, respectively, with characteristic adjustments in multiple metabolic pathways. In particular, there was a significant upregulation of amino acid, lipid, and energy metabolism. The concentrations of functional amino acids, such as glutamine, cysteine, and valine, were significantly higher in the RG group, suggesting enhanced anabolic and immune regulatory activities. Glutamine, a critical intestinal nutrient and immune modulator, has been confirmed to enhance the antioxidant capacity of fish livers and improve intestinal barrier function [55].

In terms of lipid metabolism, lysophosphatidylcholine (LPC), a bioactive phospholipid metabolite widely present in organisms, plays a key role in cell membrane structure maintenance, signal transduction, and immune regulation [56]. Glycerophospholipids, a major class that includes LPC, are also critical for maintaining liver health and resistance to oxidative damage [57,58]. Mechanistically, these compounds improve hepatic lipid accumulation by enhancing mitochondrial bioenergetic efficiency and accelerating intracellular lipid transport/degradation [59]. Their high unsaturation allows them to act as oxidative substrates, reducing reactive oxygen species (ROS) induced damage to liver membrane systems [60,61]. In this study, glycerophospholipid metabolism (as reflected by LysoPC(20:4), LysoPC(18:1), LysoPC(20:2), LysoPC(22:4), LysoPC(14:1), LysoPC(20:5), and LysoPC(16:1) levels) was upregulated in both the RG and RAG groups compared to that in the CG group, suggesting that paddy fields and recirculating aquaculture better support immune system regulation in *M. albus*. However, the RAG group exhibited lower activity in anabolic pathways (e.g., purine metabolism and amino acid metabolism) than the RG group, indicating limited growth and nutrient accumulation.

KEGG enrichment analysis revealed that the significantly enriched metabolic pathways in the RG group included amino acid biosynthesis, glycerophospholipid metabolism, and ABC transporters. These processes are closely associated with cell repair, antioxidation, and immune regulation. Glycerophospholipid metabolism is fundamental to cellular life and spans critical stages of cell growth, differentiation, and functional maintenance [62]. It also plays a pivotal role in oxidative stress responses, with the liver modulating this pathway to counteract stress-induced damage [63]. ABC transporters, which are essential for multiple physiological processes, contribute to metabolic regulation, immune defense [64], and cellular detoxification [65]. In paddy field culture, these transporters likely enhance *M. albus* health and disease resistance by improving immune function and promoting the excretion of harmful substances. These results align with the findings of Wang et al. [11], who reported that rice-eel co-culture alters metabolic profiles and boosts metabolic capacity. This phenomenon echoes prior research showing that different aquaculture environments significantly reshape fish metabolic pathways and metabolite compositions, triggering corresponding changes in physiological functions and immune regulation [66]. In this study, the upregulation of key metabolites was also strongly correlated with the functional transformations of the intestinal microbiota described earlier. For example, the increased abundance of butyrate-producing bacteria coincided with elevated SCFA levels. Therefore, we speculate that the gut-liver axis synergy in the RG group, driven by Firmicutes/Bacteroidota-mediated SCFA production from dietary fiber, underlies its improved metabolism through portal circulation-mediated hepatic modulation. Dietary fiber enrichment in the RG group promoted the proliferation of Firmicutes (notably Clostridium sensu stricto 1 and Paraclostridium) and Bacteroidetes, leading to enhanced fermentation and increased portal vein concentrations of short-chain fatty acids (SCFAs) [67,68]. Although butyrate primarily fuels colonocytes, a fraction that reaches the liver can activate PPARγ via direct receptor binding and HDAC inhibition, thereby promoting β-oxidation and glycerophospholipid synthesis, including LysoPC species [69,70]. Concurrently, SCFAs act as ligands for GPR41/43/109A, lowering colonic pH and enhancing mucus and tight junction integrity, which suppresses opportunistic pathogens, such as Plesiomonas and Aeromonas, reduces lipopolysaccharide (LPS) translocation, and alleviates hepatic inflammation [71,72]. The resultant reduction in the inflammatory burden may redirect hepatic metabolic capacity toward anabolic pathways, including glutamate and glutathione biosynthesis [73]. Finally, butyrate’s HDAC-inhibitory effect and activation of the Nrf2/PPAR pathways may upregulate hepatic ABC transporters, enhancing xenobiotic clearance [74], which is consistent with the observed increases in detoxification activity in the RG group. Future research should integrate transcriptomics and metabolic flux analysis to construct a more systematic “culture environment–microbiota–metabolism–phenotype” regulatory network model. Thus, it provides a comprehensive understanding of the multi-level interactions driving host responses to aquaculture modes.

Despite the comprehensive comparisons conducted in this study, it has several limitations. Inherent differences among farming systems—such as water quality dynamics, spatial structure, and microbial exposure—may introduce potential biases into comparative studies. Although our experimental design aimed to minimize these confounding factors by standardizing feed composition, stocking density, and management practices, some degree of systemic variation is unavoidable due to the nature of the farming models themselves. Additionally, the relatively small sample size may limit the generalizability of our findings. Future studies with larger populations and multi-site validation are warranted to confirm these results and further explore their applicability to broader aquaculture contexts.

## 5. Conclusions

In summary, the rice–fish co-culture (RG) system showed notable effects on muscle quality, intestinal microbiota composition, and liver metabolism in *Monopterus albus* when compared to conventional (CG) and recirculating aquaculture (RAG) systems. Specifically, fish reared in the RG system exhibited improved amino acid profiles and beneficial shifts in intestinal microbial communities, along with metabolic adjustments that suggest enhanced nutrient utilization and immune modulation. From an applied perspective, the RG system is a promising model for sustainable aquaculture. Its potential to improve fish quality and reduce environmental impact makes it a candidate for broader adoption; however, comprehensive evaluations of its economic feasibility and long-term performance are still required. Future studies should focus on validating key metabolic pathways, assessing growth and disease resistance, and exploring the scalability of this model in diverse aquaculture settings.

## Figures and Tables

**Figure 1 foods-14-02383-f001:**
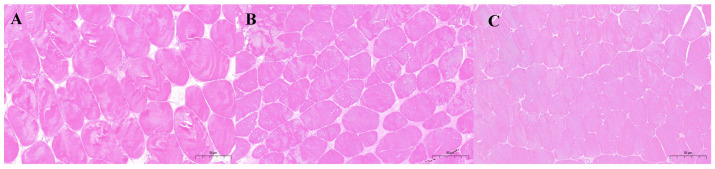
Muscle tissue structure of *M. albus* under different aquaculture modes (hematoxylin and eosin (H&E) staining). (**A**) Cage culture (CG); (**B**) Recirculating aquaculture system (RAG); (**C**) Rice-fish co-culture (RG).

**Figure 2 foods-14-02383-f002:**
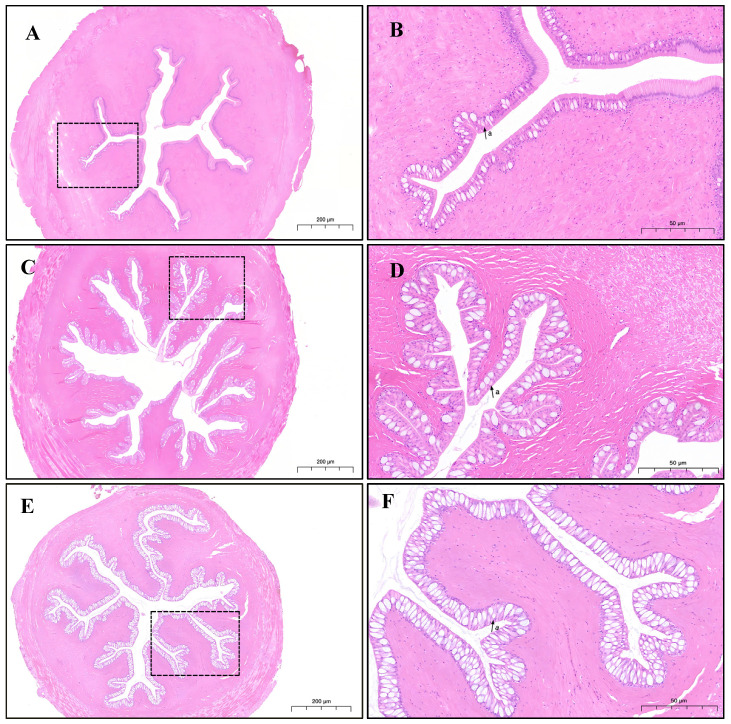
Anterior intestinal tissue morphology of *M. albus* under different aquaculture modes (HE staining). (**A**) Cage culture (CG, ×40); (**C**) recirculating aquaculture system (RAG, ×40); (**E**) rice-fish co-culture (RG, ×40); (**B**) cage culture (CG, ×100); (**D**) recirculating aquaculture system (RAG, ×100); (**F**) rice-fish co-culture (RG, ×100). (a) GC: Goblet cell.

**Figure 3 foods-14-02383-f003:**
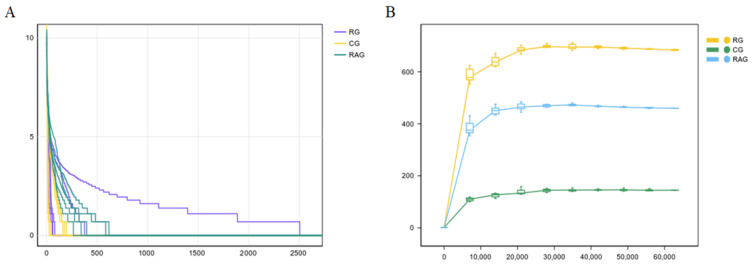
Rank-Abundance and Rarefaction Curves of *M. albus* under different farming modes. (**A**) Rank-abundance curve. (**B**) Rarefaction curves.

**Figure 4 foods-14-02383-f004:**
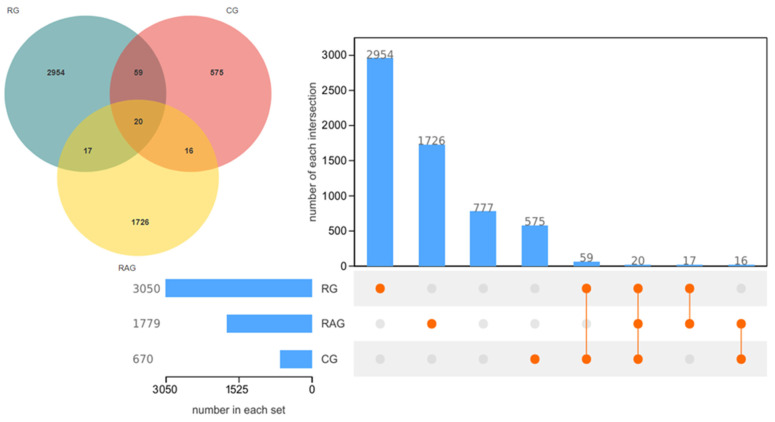
Venn diagram and operational taxonomic unit (OTU) counts of the intestinal microbiota of *M. albus* under different farming modes.

**Figure 5 foods-14-02383-f005:**
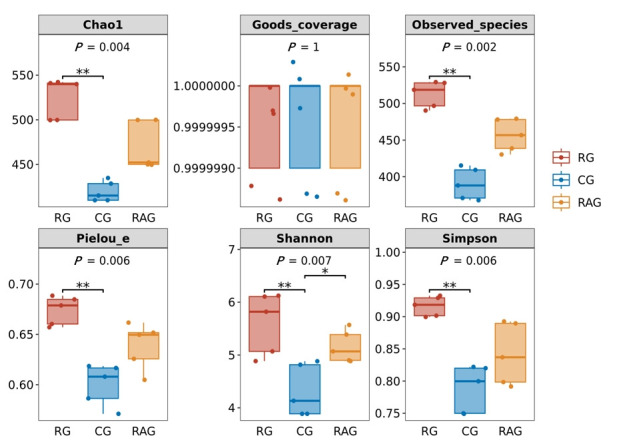
Alpha diversity analysis of the intestinal microbiota of *M. albus* under different farming modes. Values marked with asterisks indicate significant differences (* *p* < 0.05, ** *p* < 0.01).

**Figure 6 foods-14-02383-f006:**
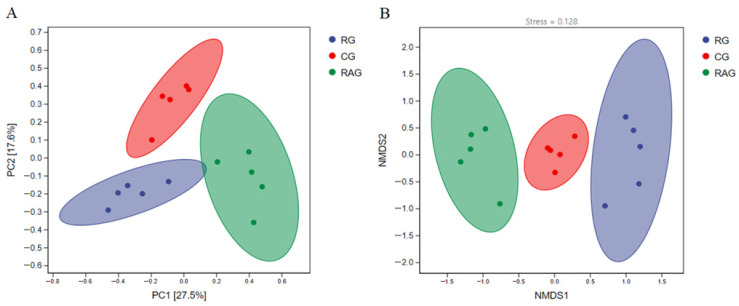
Beta diversity of the intestinal microbiota in *M. albus* under different aquaculture modes. (**A**) Principal coordinates analysis (PCoA); (**B**) non-metric multidimensional scaling (NMDS).

**Figure 7 foods-14-02383-f007:**
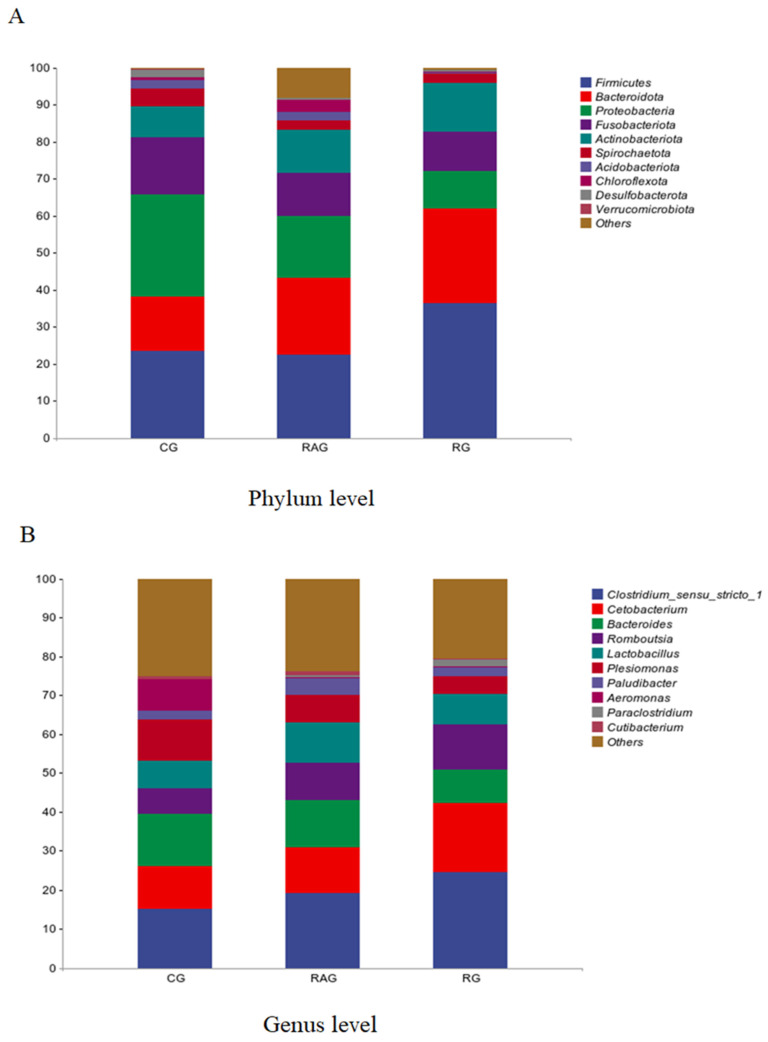
Analysis of intestinal microbiota composition of *M. albus* under different farming modes. (**A**) Average relative abundance at the phylum level; (**B**) Average relative abundance at the genus level.

**Figure 8 foods-14-02383-f008:**
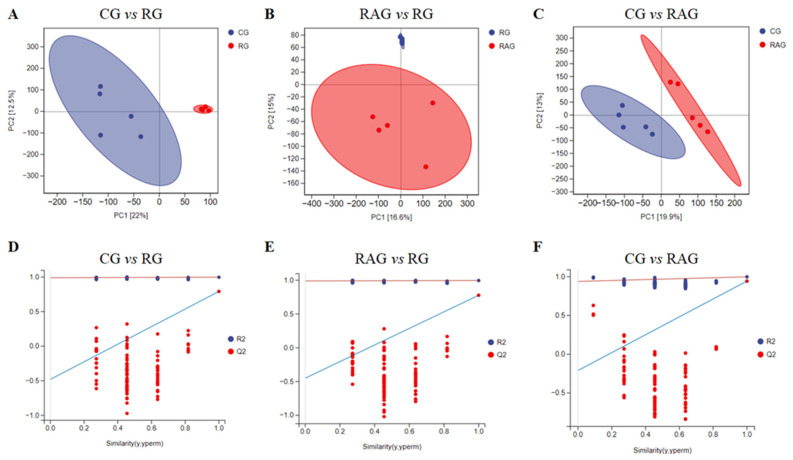
Pairwise comparative analysis of liver tissues from *M. albus* under different aquaculture modes using Principal Component Analysis (PCA) and Orthogonal Partial Least Squares Discriminant Analysis (OPLS-DA). (**A**–**C**) PCA results; (**D**–**F**) OPLS- Up, upregulated; Down, downregulated; NoDiff, no significant difference.

**Figure 9 foods-14-02383-f009:**
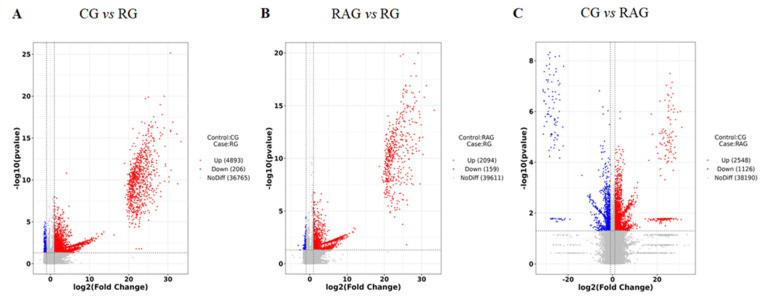
Volcano plots of differential metabolites in the liver of *M. albus* under different aquaculture modes. (**A**): CG vs. RG; (**B**): RAG vs. RG; (**C**): CG vs. RAG.

**Figure 10 foods-14-02383-f010:**
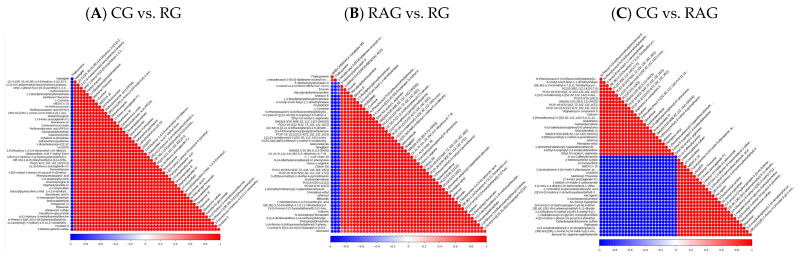
Correlation analysis of differentially abundant metabolites in the liver of *M. albus* under different aquaculture modes. (**A**) CG vs. RG; (**B**) RAG vs. RG; (**C**) CG vs. RAG.

**Figure 11 foods-14-02383-f011:**
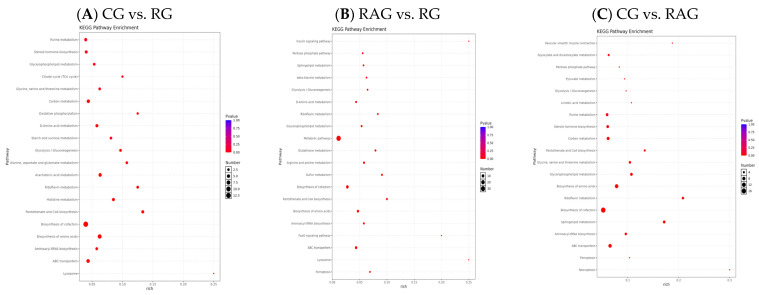
Pairwise comparison of differential metabolic pathways in the liver of *M. albus* under different aquaculture modes. (**A**) CG vs. RG; (**B**) RAG vs. RG; (**C**) CG vs. RAG. KEGG, Kyoto Encyclopedia of Genes and Genomes.

**Table 1 foods-14-02383-t001:** Elution gradients for positive and negative ion modes.

Time (min)	B %
0	5
1	5
4.7	95
6	95
6.1	5
8.5	5

**Table 2 foods-14-02383-t002:** Differences in muscle morphological indices of *M. albus* under different farming modes.

Index	CG	RAG	RG
Number of muscle fibers	43.00 ± 5.23 ^a^	48.13 ± 5.17 ^ab^	57.40 ± 6.21 ^b^
Density of muscle fibers (mm^2^)	96.38 ± 11.02 ^a^	101.76 ± 10.22 ^a^	134.61 ± 13.02 ^b^

**Note:** Values are presented as mean ± SD (standard deviation). Superscript letters in the same row represent statistically significant differences (*p* < 0.05).

**Table 3 foods-14-02383-t003:** Muscle proximate content (g/kg, wet weight) under different farming modes of *M. albus*.

Index	CG	RAG	RG
Moisture	752.85 ± 5.17 ^a^	724.31 ± 7.58 ^b^	696.07 ± 5.21 ^c^
Crude protein	205.11 ± 6.17 ^a^	225.50 ± 6.82 ^b^	239.81 ± 6.38 ^b^
Crude lipid	17.04 ± 2.03 ^a^	14.51 ± 2.11 ^ab^	10.07 ± 3.41 ^b^
Crude ash	18.88 ± 2.07 ^a^	13.56 ± 2.04 ^b^	13.86 ± 2.55 ^b^

**Note:** Values are presented as mean ± standard deviation (SD). Superscript letters in the same row represent statistically significant differences (*p* < 0.05).

**Table 4 foods-14-02383-t004:** Amino acid content in muscle (g/kg dry matter) under different farming modes of *M. albus*.

Index	CG	RAG	RG
Aspartate	7.53 ± 0.30 ^a^	9.94 ± 0.66 ^b^	10.62 ± 0.60 ^b^
Glutamate	16.79 ± 0.67 ^a^	17.35 ± 0.57 ^b^	19.26 ± 0.39 ^b^
Glycine	3.45 ± 0.24 ^a^	4.13 ± 0.35 ^b^	4.43 ± 0.45 ^b^
Phenylalanine	6.36 ± 0.34 ^a^	6.82 ± 0.44 ^ab^	7.16 ± 0.31 ^b^
Tyrosine	3.83 ± 0.12	3.79 ± 0.54	4.36 ± 0.39
Alanine	5.12 ± 0.52	5.50 ± 0.28	5.57 ± 0.34
Serine	4.02 ± 0.38	4.14 ± 0.32	4.50 ± 0.23
Histidine	29.98 ± 0.43 ^a^	32.39 ± 0.25 ^b^	33.97 ± 0.64 ^c^
Threonine	1.70 ± 0.46 ^a^	2.42 ± 0.17 ^ab^	2.95 ± 1.24 ^b^
Arginine	10.26 ± 0.52 ^a^	10.59 ± 0.38 ^a^	12.19 ± 0.61 ^b^
Cysteine	0.99 ± 0.44	1.15 ± 0.41	1.25 ± 0.48
Valine	4.24 ± 0.33 ^a^	5.01 ± 0.63 ^b^	5.37 ± 0.32 ^b^
Methionine	3.43 ± 0.23	3.78 ± 0.46	4.99 ± 0.44
Isoleucine	5.28 ± 0.41 ^a^	5.48 ± 0.61 ^a^	6.37 ± 0.52 ^b^
Leucine	8.60 ± 0.63 ^a^	9.98 ± 0.66 ^b^	10.05 ± 0.24 ^b^
Lysine	14.92 ± 0.45 ^a^	15.82 ± 0.53 ^b^	16.60 ± 0.44 ^b^
Proline	8.50 ± 0.40	8.77 ± 0.33	8.88 ± 0.53
TUAAs	43.12 ± 0.91 ^a^	47.54 ± 0.68 ^b^	51.39 ± 1.14 ^c^
TNEAAs	90.50 ± 2.31 ^a^	97.77 ± 0.64 ^b^	105.04 ± 1.94 ^c^
TEAAs	44.53 ± 0.57 ^a^	49.31 ± 0.85 ^b^	52.49 ± 1.50 ^c^
TAAs	135.04 ± 2.11 ^a^	147.07 ± 1.39 ^b^	157.53 ± 2.61 ^c^

**Note:** Values are presented as mean ± SD (standard deviation). Superscript letters in the same row represent statistically significant differences (*p* < 0.05). **Abbreviations:** TUAA, total umami amino acids; TNEAA, total non-essential amino acids; TEAA, total essential amino acids; TAA, total amino acids.

**Table 5 foods-14-02383-t005:** Fatty acid profile (percentage of fatty acids, %) under different farming modes of *M. albus*.

Index	CG	RAG	RG
Undecanoic acid (C11:0)	0.66 ± 0.16 ^a^	0.82 ± 0.08 ^b^	1.04 ± 0.05 ^c^
Lauric acid (C12:0)	1.26 ± 0.11 ^a^	1.37 ± 0.11 ^ab^	1.57 ± 0.24 ^b^
Tridecanoic acid (C13:0)	0.30 ± 0.13	0.31 ± 0.19	0.33 ± 0.07
Myristic acid (C14:0)	3.24 ± 0.10 ^c^	2.77 ± 0.39 ^b^	2.14 ± 0.26 ^a^
Pentadecanoic acid (C15:0)	1.18 ± 0.14 ^a^	1.31 ± 0.12 ^ab^	1.42 ± 0.13 ^b^
Palmitic acid (C16:0)	23.17 ± 0.64 ^a^	23.67 ± 0.61 ^b^	25.10 ± 0.13 ^c^
Stearic acid (C18:0)	4.16 ± 0.29	4.26 ± 0.21	4.25 ± 0.12
Arachidonic acid (C20:0)	0.41 ± 0.04	0.49 ± 0.07	0.33 ± 0.20
Total saturated fatty acids (SFA)	34.38 ± 0.83 ^a^	35.01 ± 0.62 ^b^	36.18 ± 0.33 ^c^
Palmitoleic acid (C16:1)	9.20 ± 0.11 ^a^	9.65 ± 0.25 ^b^	10.42 ± 0.29 ^c^
Cis-10-Heptadecenoic acid (C17:1)	0.76 ± 0.08 ^a^	0.81 ± 0.09 ^a^	1.10 ± 0.25 ^c^
Oleic acid (C18:1)	24.65 ± 1.01 ^a^	24.31 ± 0.24 ^a^	25.43 ± 0.31 ^b^
Nervonic acid (C24:1)	1.40 ± 0.11	1.45 ± 0.09	1.46 ± 0.06
Total monounsaturated fatty acids (MUFA)	36.00 ± 1.01 ^a^	36.21 ± 0.46 ^a^	38.42 ± 0.31 ^b^
Linoleic acid (C18:2)	21.40 ± 0.70 ^c^	20.47 ± 0.34 ^b^	17.07 ± 0.69 ^a^
Arachidonic acid (C20:4)	2.25 ± 0.17 ^a^	2.19 ± 0.10 ^a^	1.30 ± 0.33 ^b^
n-6 polyunsaturated fatty acids (n-6 PUFA)	23.64 ± 0.82 ^c^	22.66 ± 0.37 ^b^	18.36 ± 0.98 ^a^
Linolenic acid (C18:3)	2.17 ± 0.12 ^a^	2.29 ± 0.07 ^a^	2.57 ± 0.18 ^b^
Eicosapentaenoic acid (C20:5)	2.22 ± 0.11 ^a^	2.29 ± 0.04 ^a^	2.56 ± 0.31 ^b^
Docosahexaenoic acid (C22:6)	1.04 ± 0.20 ^a^	1.05 ± 0.54 ^ab^	1.91 ± 0.61 ^b^
n-3 polyunsaturated fatty acids (n-3 PUFA)	5.42 ± 0.35 ^a^	5.63 ± 0.54 ^a^	7.04 ± 0.50 ^b^
Total polyunsaturated fatty acids (PUFA)	29.06 ± 0.60 ^b^	28.29 ± 0.59 ^b^	25.40 ± 0.72 ^a^
n-3 PUFA/ n-6 PUFA	0.23 ± 0.02 ^a^	0.25 ± 0.02 ^a^	0.38 ± 0.03 ^b^

**Note:** Values are presented as mean ± SD (standard deviation). Superscript letters in the same row represent statistically significant differences (*p* < 0.05). **Abbreviations:** SFA, total saturated fatty acids; MUFA, total monounsaturated fatty acids; PUFA, total polyunsaturated fatty acids; n-3 PUFA, total n-3 polyunsaturated fatty acids; n-6 PUFA, total n-6 polyunsaturated fatty acids.

**Table 6 foods-14-02383-t006:** Intestinal morphology of M. albus under different farming modes.

Index	CG	RAG	RG
Villus circumferences (mm)	8653.6 ± 421.1 ^a^	11,009.4 ± 417.5 ^b^	11,094.1 ± 324.7 ^b^
Goblet cell amounts (A/root)	84.2 ± 13.2 ^a^	121.5 ± 16.7 ^b^	205.4 ± 13.6 ^c^

**Note:** Values are presented as mean ± SD (standard deviation). Superscript letters in the same row represent statistically significant differences (*p* < 0.05).

**Table 7 foods-14-02383-t007:** Dominant bacteria and their relative abundances (%) in the intestinal microbiota of *M. albus* under different farming modes.

Index	CG	RAG	RG
**Phylum Level**			
Firmicutes	23.49 ± 3.12 ^a^	22.58 ± 9.02 ^a^	36.49 ± 3.79 ^b^
Bacteroidota	14.71 ± 7.14 ^a^	19.66 ± 1.95 ^a^	25.50 ± 2.29 ^b^
Proteobacteria	27.66 ± 8.89 ^b^	16.86 ± 4.26 ^a^	10.03 ± 5.87 ^a^
Fusobacteriota	15.34 ± 6.32	11.60 ± 5.80	10.66 ± 0.58
Actinobacteriota	8.31 ± 5.04	11.65 ± 1.04	13.19 ± 3.61
Spirochaetota	4.89 ± 2.20	2.55 ± 1.11	2.34 ± 1.14
Acidobacteriota	2.23 ± 1.95	2.24 ± 1.22	0.21 ± 0.13
Chloroflexota	0.79 ± 2.38	3.34 ± 4.37	0.31 ± 0.29
Desulfobacterota	2.19 ± 1.60	0.43 ± 0.66	0.56 ± 0.48
Verrucomicrobiota	0.13 ± 0.57	0.01 ± 0.42	0.07 ± 0.78
Others	0.27 ± 0.45	8.10 ± 13.90	0.21 ± 0.38
**Genus level**			
*Clostridium_sensu_stricto_1*	15.17 ± 0.80 ^a^	19.16 ± 0.47 ^b^	24.43 ± 0.91 ^c^
*Cetobacterium*	10.82 ± 1.39 ^a^	11.60 ± 1.28 ^a^	17.77 ± 1.13 ^b^
*Bacteroides*	13.59 ± 2.91 ^b^	12.31 ± 2.39 ^ab^	8.71 ± 2.22 ^a^
*Romboutsia*	6.38 ± 2.58 ^a^	9.55 ± 0.59 ^b^	11.52 ± 1.51 ^b^
*Lactobacillus*	7.22 ± 3.07	10.33 ± 1.53	8.03 ± 3.05
*Plesiomonas*	10.63 ± 0.84 ^c^	7.21 ± 2.10 ^b^	4.53 ± 1.60 ^a^
*Paludibacter*	2.35 ± 1.50	4.18 ± 3.24	2.29 ± 0.64
*Aeromonas*	7.99 ± 2.44 ^b^	0.21 ± 0.56 ^a^	0.29 ± 0.25 ^a^
*Paraclostridium*	0.10 ± 0.11 ^a^	0.61 ± 1.41 ^a^	1.78 ± 0.55 ^b^
*Cutibacterium*	0.74 ± 0.73	0.98 ± 1.15	0.18 ± 0.22
Others	25.02 ± 5.07	23.86 ± 0.88	20.48 ± 4.77

**Note:** Values are presented as mean ± SD (standard deviation). Superscript letters in the same row represent statistically significant differences (*p* < 0.05).

## Data Availability

The original contributions presented in this study are included in the article and Appendix A. Further inquiries can be directed to the corresponding author.

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
