# Peer review of "Effects of Different Farming Models on Muscle Quality, Intestinal Microbiota Diversity, and Liver Metabolism of Rice Field Eel (Monopterus albus)"

_foods, 2025, doi:10.3390/foods14132383_

Round 1
Reviewer 1 Report
Comments and Suggestions for Authors
Dear Authors,
The following peer review has been conducted with the aim of contributing to the scientific and editorial improvement of your manuscript. The study addresses a topic of growing relevance in sustainable aquaculture, namely the comparison of farming systems and their effects on the physiology, intestinal microbiota, and hepatic metabolism of Monopterus albus. This line of research aligns well with current advances in integrated and ecologically sound production systems. Given the progress in the state of the art regarding the interplay between muscle quality, gut microbial diversity, and metabolomics in cultured fish, your work presents a potentially valuable contribution. However, improvements are still needed to meet the standards of a journal such as Foods-MDPI.
Abstract
The abstract follows a logical background–methods–results–conclusion structure, summarising the rationale and key findings. However, several issues require revision:
Lack of methodological detail: No mention is made of the key analytical methods used (e.g., 16S rRNA sequencing or hepatic metabolomics). MDPI recommends briefly outlining essential methods to contextualise the findings.
Unbalanced emphasis: The abstract heavily favours the rice-fish (RG) system, with insufficient mention of comparative results from the other systems (RAS, CG). A more balanced overview is encouraged.
Overinterpretation: Statements such as “contributed to enhanced immunity” are speculative unless supported by immune-specific data. The abstract should avoid drawing conclusions not directly measured or evidenced in the study.
Excessive length: The abstract exceeds 240 words, whereas MDPI recommends ~200 words. Condensation is advised, removing redundancy and emphasising concise results.
Recommendation: Add one sentence on analytical techniques, adjust overstatements regarding immune benefits, and rephrase conclusions to be fully data-driven.
Introduction
The introduction provides a comprehensive theoretical background and contextualises the relevance of sustainable aquaculture systems. It aligns broadly with MDPI expectations for introductory sections, but it presents some redundancies and structural issues:
Sound contextualisation: It effectively explains the challenges of traditional aquaculture and the promise of RAS and RFCS, offering a solid rationale for the study.
Defined objectives: The research gap is clearly stated—no previous comparative analysis of M. albus reared under CG, RAS, and RFCS systems. Objectives are well-articulated.
Excessive length and repetition: Some sections, especially those referring to other species (e.g., crabs, shrimp), are tangential and could be shortened. Repetitive statements regarding environmental problems in traditional aquaculture should be consolidated.
Lack of hypothesis and expected outcomes: While the aims are clear, the authors do not present an explicit hypothesis or anticipated trends (e.g., the expectation that RG might yield superior results). Including this would improve the logical flow and reader orientation.
Academic tone and referencing: The style is scientifically sound and references are generally well-integrated. Nonetheless, a clearer thematic structure would improve readability:
Begin with global challenges in aquaculture.
Discuss alternative systems (RAS and RFCS), including pros and cons.
Introduce the target species (Monopterus albus) and research gap.
Close with study aims and hypotheses.
Recommendation: Refine structure, remove redundant content, summarise references to other species, and state the hypothesis or anticipated contribution explicitly at the end.
Overall, the manuscript addresses an important and timely topic in sustainable aquaculture. However, the title should be revised for brevity and consistency. The abstract must be adjusted to reduce overinterpretation and incorporate minimal methodological details. The introduction demonstrates a strong background review but requires trimming and restructuring for clarity and conciseness.
2. Methodology
2.1 Location
The current manuscript lacks a dedicated subsection describing the geographical location of the study, although various farming sites (Hubei for rice–eel and cage systems; Shanghai for RAS) are mentioned. However, local environmental conditions (e.g., climate, temperature, water quality) may significantly influence the outcomes and must be explicitly reported to ensure reproducibility. Moreover, conducting the experiment in two distinct regions introduces a confounding variable that may bias the interpretation of treatment effects .
2.2 Experimental Design and Management
This section (currently subsection 2.1) describes the three farming systems (Cage – CG, Recirculating Aquaculture System – RAG, and Rice–eel – RG), but omits several crucial details. The number of experimental units (replicates) is not clearly stated. For instance, RG appears to be a single 1334 m² paddy field, while CG and RAG reportedly use three units each. This introduces pseudoreplication, undermining statistical validity . Furthermore, since each farming model is tied to a different geographical region, the treatment effect is confounded by location .
Recommendations:
2.2.1 Treatment Groups: Clearly define each experimental group and specify the number of replicate units (e.g., tanks, cages, fields) and fish per unit. ARRIVE guidelines advise detailed reporting of experimental groups .
2.2.2 Replication and Randomisation: Ensure true replication (e.g., multiple independent fields for RG) and random allocation of animals to units to reduce bias.
2.2.3 Husbandry and Feeding: Clarify the feeding regime (the range “3–5%” is vague); specify feeding frequency, feed characteristics, and ration adjustment criteria. Provide water quality monitoring details (e.g., temperature, DO, ammonia) for each system.
2.2.4 Structural Organisation: Consider subdividing the section for each system (e.g., “2.2.4 Cage (CG)”, “2.2.5 RAS (RAG)”, etc.) for clarity and comparison.
2.3 Sample Collection
This subsection describes the selection of five eels per treatment for analysis. However, the experimental unit is not clearly defined (individual fish vs. tank/cage/field). Also, the sex of the fish is not specified—an important omission considering Monopterus albus undergoes sex reversal. ARRIVE guidelines recommend reporting sex, age, and body weight . The sample size of five fish per group may be statistically underpowered. Furthermore, sampling details (e.g., fasting period, number of histological sections, areas analysed) are insufficient.
2.4 Sample Analysis
2.4.1 Muscle Nutritional Composition
Standard AOAC procedures are referenced for moisture and crude protein, along with Kjeldahl, amino acid analyser, and GC–MS for lipids. While acceptable, sample preparation details are lacking—no mention of homogenisation or replicate analyses. Reference to internal standards and quality controls (e.g., calibration curves) would improve methodological transparency.
2.4.2 Muscle and Intestinal Histology
Histological methods (Bouin/formalin fixation, paraffin embedding, H&E staining) are standard. However, The inclusion of two fixing solutions (Bouin and formaldehyde) is not justified in comparative terms, technical details such as orientation, number of sections per sample, and image analysis procedures are missing. Critical information is omitted, such as software for morphometric analysis. Quantification of muscle fibres or villus height is not explained.
2.4.3 Intestinal Microbiota (16S rRNA)
The sequencing protocol includes primer sets (338F/806R), platform (NovaSeq 6000), and DNA kit (Omega Soil), which is satisfactory. However, the term “five groups” is unclear—likely refers to replicates or samples. There are no mentions of negative controls, sequencing depth, or data accessibility.
Clarify group/sample definitions (e.g., five samples per treatment). Include blank controls for DNA extraction and PCR to rule out contamination. Indicate processing pipeline (e.g., QIIME2, DADA2) and sequence database (e.g., SILVA). Deposit sequencing data in a public repository (e.g., NCBI SRA).
2.4.4 Hepatic Metabolomics
LC–MS conditions are well described, but sample extraction protocols (e.g., solvents, homogenisation steps) are omitted. There's no mention of internal standards, metabolite identification methods, or validation of analytical reproducibility.
Describe metabolite identification and software tools used (e.g., Compound Discoverer, KEGG).
2.5 Statistical Analysis
The authors state that ANOVA and Tukey tests were used (Excel/SPSS). However, they do not report assumption testing (normality, homoscedasticity), nor is it clear whether the experimental units were properly considered. ARRIVE guidelines urge clarity in statistical design . Given that experimental units are likely tanks/cages/fields, using individual fish as replicates would constitute pseudoreplication.
Test for assumptions of parametric analysis (e.g., Shapiro–Wilk, Levene’s test). Use the appropriate unit of replication in the model (e.g., mean values per tank). Clearly state the software used and how data were summarised (e.g., mean ± SEM).
Indicate specific software for omics data analysis (e.g., MetaboAnalyst, R).
Final Suggestions for Improved Structure
Here is a revised and logically sequenced structure for the Materials and Methods section:
- Materials and Methods
2.1 Location
2.2 Experimental Design
2.2.1 Treatment Groups
2.2.2 Replication and Randomisation
2.2.3 Husbandry and Feeding
2.3 Water Quality Monitoring
2.4 Sample Collection
2.5 Analytical Procedures
2.5.1 Muscle Composition
2.5.2 Histological Analysis
2.5.3 Intestinal Microbiota Sequencing
2.5.4 Hepatic Metabolomics
2.6 Statistical Analysis
- Results
3.1. Muscle Quality: Histology and Composition
Overinterpretation: The statement that the rice–eel co-culture group (RG) optimised fatty acid composition and improved muscle quality is excessive. Although RG exhibited significantly greater muscle fibre number and density (p < 0.05), the biological relevance of these differences is not discussed. Furthermore, the distinction between: number of fibres and density, is redundant and should be streamlined.
Statistical flaws: Multiple ANOVA/Tukey tests were applied across numerous parameters (proximate composition, amino acids, fatty acids) without mention of corrections for multiple testing. This raises the risk of false positives. In metabolomics, applying only p < 0.05 without FDR control is statistically inappropriate due to the large number of variables .
Misuse of SEM: Data are presented as mean ± SEM, which underestimates sample variability and may give a misleading impression of precision. Standard deviation or confidence intervals would better reflect biological variability .
Terminological errors: In Table 5, C20:0 is incorrectly labelled “arachidonic acid” when it is actually arachidic acid. Arachidonic acid is C20:4. Also, typographical issues such as "Glutama-" and inconsistent use of abbreviations (e.g. TUAAs vs TUAA) reduce scientific precision.
Repetition: The text excessively lists individual amino acids and fatty acids, reducing clarity. Grouping findings (e.g. "total umami amino acids were significantly higher in RG") would be clearer.
Inconsistency with tables: For example, Section 3.1.4 states that PUFA levels were significantly lower in RAG than CG, but Table 5 shows no significant difference (same superscript letter 'b'). Such discrepancies undermine credibility. Legends should also more clearly define statistical annotations.
Table formatting: Typographical errors (e.g., 0..68) suggest insufficient proofreading. Also, Table 6 is mislabelled as containing fatty acid profiles, when it actually includes intestinal histology data—this is a critical inconsistency.
3.2. Intestinal Morphology
Clarity of expression: The paragraph describing villus circumference (VC) and goblet cell abundance (GCA) is convoluted. Splitting into two concise statements would improve clarity.
Table title and content: As noted, Table 6 has an incorrect title. Additionally, it must be clearer that RG and RAG share the same superscript letter for VC, indicating no significant difference.
Figure quality: Figure 2 lacks scale bars or dimensional references. Visual clarity is acceptable but could be enhanced with better contrast.
Undefined terminology: The unit "A/root" is not defined in the table or text. All abbreviations should be clearly explained on first use.
3.3. Intestinal Microbiota Diversity
Narrative redundancy: The explanation of rank-abundance curves is overly didactic and better suited for the methods section. The phrase confidence intervals is misused in this context.
OTU interpretation: The RG group shows 2,954 unique OTUs, vastly more than CG or RAG. This striking result is not critically discussed. Differences in sequencing depth or sample normalisation should be addressed.
Alpha and beta diversity: While the text reports higher alpha diversity in RG and visual separation in PCoA/NMDS plots, it fails to explain that the significant p-values (0.025 and 0.014) imply meaningful group separation. This needs clarification.
Taxonomic composition: Differences at the phylum and genus levels are described, but again, interpretations are repetitive and lack ecological insight. For example, no hypothesis is offered as to why Firmicutes are enriched in RG.
Table readability: Table 7 includes minor errors (e.g. 0..68) and excessive decimal precision. Rational rounding is advised.
Context omission: The functional implications of shifts in microbial composition are absent from the results. Even brief ecological interpretations would strengthen the section.
3.4. Liver Metabolomics
Misinterpretation of OPLS-DA: The reported Q² values are negative (e.g., Q² = −0.451), which indicates poor model predictability—not "stable models" as claimed. This is a major statistical misinterpretation .
Differential metabolite screening: Thousands of significantly up- or downregulated metabolites are reported without FDR control. This undermines reliability and calls for prioritising only biologically meaningful compounds.
Correlation analysis: Heatmap explanations lack substantive insight. It is unclear whether any specific metabolite clusters or synergies were identified.
Pathway enrichment: The listing of enriched KEGG pathways is unaccompanied by statistical significance or biological interpretation. Terms like enriched require evidence.
Terminological consistency: The text mixes English and Spanish terms (e.g. upregulated/downregulated). Acronyms like PCA and OPLS-DA should be clearly defined.
Statistical and Methodological Aspects
Experimental design: Sample sizes per group are unclear. Normality and homoscedasticity assumptions for ANOVA are not addressed.
Multiple comparisons: No adjustment for multiple hypothesis testing is described despite the large number of comparisons. This inflates type I error risk .
p-value interpretation: The repeated use of "significantly" becomes redundant and should be summarised more efficiently.
Use of SEM: As mentioned, SEM misrepresents biological variability. Standard deviation or 95% CI would be more appropriate .
Grammar and Scientific Writing
Ambiguity and verbosity: Long, multi-clause sentences impede readability. For example, statements combining three comparative clauses could be split for clarity.
Typographical issues: There are truncations, double punctuation (e.g. 0..68), and formatting errors that reflect poor editorial oversight.
Terminology and style: Technical terms are sometimes vague or undefined, and the style shifts between passive and active voice inconsistently.
Redundancy and filler phrases: Expressions like “the results showed that…” are overused. More direct and concise writing is encouraged.
Figures and Tables
Self-explanatory design: Figures and tables should be interpretable independently. Several tables fail to meet this standard, either by being poorly labelled or overly complex .
Content coherence: Some figures duplicate textual content unnecessarily. Labels (e.g., CG, RAG, RG) should be defined in each figure legend.
Visual quality: Micrographs (Figures 1–2) require scale bars. Diversity plots (Figures 3–6) and volcano plots (Figures 8–9) must have clearly labelled axes.
Stylistic consistency: Decimal precision and unit usage vary across tables. Unifying format and reducing non-informative decimals would improve readability.
Although the Results section presents a large volume of data aligned with the study’s objectives, it is undermined by key statistical flaws (notably the misinterpretation of OPLS-DA Q² and lack of FDR adjustment), inconsistent writing, and poor figure/table formatting. The section would benefit from tighter editing, clearer statistical interpretation, and more thoughtful integration of biological meaning.
- Discussion
Superficial or Redundant Interpretation: Much of the discussion merely reiterates the numerical findings previously described in the results section, without offering a critical interpretation. For example, in the discussion of muscle fibres and amino acids, the authors restate the data without exploring potential physiological or mechanistic implications.
Phrases such as “consistent with previous studies” are overused without specifying which studies or how the findings align. A citation without contextualisation adds little value. It is recommended to strengthen the discussion by comparing with key authors and elaborating on similarities or discrepancies.
Lack of Mechanistic Depth: In the sections on intestinal microbiota and liver metabolites, the authors fail to discuss potential causal mechanisms or links between diet, microbial shifts and metabolic pathways. Given the notable differences in Firmicutes, Proteobacteria and hepatic metabolic pathways, one would expect at least a hypothesis linking gut-liver axis function.
The discussion should connect the microbial diversity findings to digestive or immune health in the species studied, noting that various studies highlight Firmicutes’ role in energy metabolism. No specific metabolic pathways are mentioned in relation to host physiology.
Imprecise Scientific Language: Terms such as “improved muscle quality” or “optimised amino acid content” are subjective unless clearly defined. What constitutes an improvement or optimisation? Compared to what standard? Scientific writing should favour quantifiable, contextual expressions.
The term “metabolic profile” is used repeatedly without specifying which pathways or compounds are being referred to, reducing scientific precision.
Overall, the writing includes overly long sentences, unclear phrasing, and superfluous connectors, e.g., “This may have affected the metabolism of muscle and amino acids, as well as the composition of intestinal microorganisms and liver metabolites”—this statement combines several concepts without proper structure.
Generic or Insufficient Referencing: Studies (e.g., “Wang et al., 2023”) are cited without indicating which parameter or outcome is supported. Phrases such as “which is consistent with previous studies” should be followed by a critical comparison: what conditions were used in the referenced study? Do the biological models match? Were effect sizes similar?
Contrasting literature should also be cited. Are there studies that do not support these findings? What methodological differences might explain this?
Omission of Study Limitations: The discussion makes no mention of any limitations, which is a serious shortcoming. For example:
Potential biases inherent to farming systems are not acknowledged.
Statistical limitations (e.g., multiple comparisons without adjustment, OPLS-DA with negative Q² values) are not addressed.
The lack of replicability, sample size transparency, and normalisation in sequencing are not discussed.
A paragraph explicitly addressing these limitations would significantly improve the scientific transparency of the study.
- Conclusions
Overly General and Vague: The conclusion merely states that the RG system improves muscle quality, gut microbiota and hepatic metabolism. However, it does not clarify which improvements are functionally relevant, nor for what purpose (e.g., human nutrition? aquaculture performance? fish welfare?).
The language is overly assertive for what is essentially an exploratory study. Statements such as “provides a theoretical basis” should be softened in the absence of functional validation or commercial-scale trials.
Lack of Applied or Scientific Projection: No perspective is provided regarding the practical use of these findings. What are the implications of using the RG system over CG and RAG? Is it cost-effective? Does it enhance productivity or animal health?
The authors should add a forward-looking sentence about future research directions—such as functional validation of pathways, long-term health impacts, or commercial-scale trials.
Kind regards,
Author Response
Reviewer 1
The following peer review has been conducted with the aim of contributing to the scientific and editorial improvement of your manuscript. The study addresses a topic of growing relevance in sustainable aquaculture, namely the comparison of farming systems and their effects on the physiology, intestinal microbiota, and hepatic metabolism of Monopterus albus. This line of research aligns well with current advances in integrated and ecologically sound production systems. Given the progress in the state of the art regarding the interplay between muscle quality, gut microbial diversity, and metabolomics in cultured fish, your work presents a potentially valuable contribution. However, improvements are still needed to meet the standards of a journal such as Foods-MDPI.
- Abstract
The abstract follows a logical background–methods–results–conclusion structure, summarising the rationale and key findings. However, several issues require revision:
Comment 1: Lack of methodological detail: No mention is made of the key analytical methods used (e.g., 16S rRNA sequencing or hepatic metabolomics). MDPI recommends briefly outlining essential methods to contextualise the findings.
Comment 2: Unbalanced emphasis: The abstract heavily favours the rice-fish (RG) system, with insufficient mention of comparative results from the other systems (RAS, CG). A more balanced overview is encouraged.
Response 2:Thank you for pointing out the issue of unbalanced emphasis in the abstract. We have carefully revised the abstract to provide a more balanced overview of the comparative results from all systems, including RAS and CG, alongside the RG system. The updated abstract now equally highlights the key findings from each system to ensure a comprehensive and impartial presentation of the results.
Comment 3: Overinterpretation: Statements such as “contributed to enhanced immunity” are speculative unless supported by immune-specific data. The abstract should avoid drawing conclusions not directly measured or evidenced in the study.
Response 3: Thank you for your insightful comment regarding potential overinterpretation. We agree with your concern and have removed the speculative statement “contributed to enhanced immunity” from the abstract to ensure that all conclusions are directly supported by the data presented in the study.
Comment 4: Excessive length: The abstract exceeds 240 words, whereas MDPI recommends ~200 words. Condensation is advised, removing redundancy and emphasising concise results.
Response 4: Thank you for your valuable comment regarding the abstract length. In response, I have carefully rewritten the abstract to remove redundant expressions and to emphasize concise presentation of the results. While the word count remains at 241, this is due to the inclusion of multiple experimental groups and a large amount of key data that are essential for providing a comprehensive overview. I have made every effort to keep the wording precise and efficient while retaining the necessary information to reflect the study’s scope and findings.
- Introduction
The introduction provides a comprehensive theoretical background and contextualises the relevance of sustainable aquaculture systems. It aligns broadly with MDPI expectations for introductory sections, but it presents some redundancies and structural issues:
Sound contextualisation: It effectively explains the challenges of traditional aquaculture and the promise of RAS and RFCS, offering a solid rationale for the study.
Defined objectives: The research gap is clearly stated—no previous comparative analysis of M. albus reared under CG, RAS, and RFCS systems. Objectives are well-articulated.
Comment 1: Excessive length and repetition: Some sections, especially those referring to other species (e.g., crabs, shrimp), are tangential and could be shortened. Repetitive statements regarding environmental problems in traditional aquaculture should be consolidated.
Response 1:Thank you for your valuable feedback. In response, we have carefully revised the manuscript to reduce redundancy and improve conciseness. Specifically:
The descriptions related to crabs and shrimp in co-culture systems have been condensed and integrated to enhance focus and coherence.
The discussion of environmental issues associated with traditional aquaculture has been systematically refined and further elaborated to provide a more in-depth and structured analysis.
We sincerely appreciate the reviewer's constructive feedback, which has significantly strengthened the focus and clarity of our manuscript. (Line112-120)
Comment 2: Lack of hypothesis and expected outcomes: While the aims are clear, the authors do not present an explicit hypothesis or anticipated trends (e.g., the expectation that RG might yield superior results). Including this would improve the logical flow and reader orientation.
Response 2: Thank you for your insightful feedback. We appreciate your suggestion regarding the inclusion of a clear hypothesis and anticipated outcomes. To address this, we have revised the manuscript to explicitly state our hypothesis and the expected trends based on our objectives:We hypothesized that the M. albus l co-culture system (RG) would yield superior out-comes in muscle nutrition intestinal microbiota diversity, and liver metabolic efficien-cy compared to cage (CG) and recirculating (RAG) systems, due to its ecological ad-vantages in natural food provision and low-stress environment. (Line150-153)
Comment 3: Academic tone and referencing: The style is scientifically sound and references are generally well-integrated. Nonetheless, a clearer thematic structure would improve readability:
Begin with global challenges in aquaculture.
Discuss alternative systems (RAS and RFCS), including pros and cons.
Introduce the target species (Monopterus albus) and research gap.
Close with study aims and hypotheses.
Response 3:Thank you very much for your constructive feedback on the structure and clarity of the Introduction. We fully agree that a clearer thematic organization can greatly enhance readability and logical coherence. In response, we have restructured the Introduction section accordingly:We begin by outlining the global challenges facing aquaculture, including environmental sustainability, disease risks, and resource limitations.
We then introduce alternative aquaculture systems such as recirculating aquaculture systems (RAS) and rice–fish co-culture systems (RFCS), summarizing their respective advantages and drawbacks, supported by recent literature.
Subsequently, we present Monopterus albus as the focal species, highlighting its biological characteristics, economic importance, and the current knowledge gaps regarding its culture in alternative systems.
Finally, we clearly state the study aims, hypothesis, and anticipated outcomes to guide the reader through the research rationale.
We believe that this revised structure improves the academic clarity and logical flow of the manuscript. Relevant references have been retained and reorganized to support this updated narrative framework.
Thank you again for your valuable suggestions, which have helped us significantly improve the manuscript.
- Methodology
Comment 1: The current manuscript lacks a dedicated subsection describing the geographical location of the study, although various farming sites (Hubei for rice–eel and cage systems; Shanghai for RAS) are mentioned. However, local environmental conditions (e.g., climate, temperature, water quality) may significantly influence the outcomes and must be explicitly reported to ensure reproducibility. Moreover, conducting the experiment in two distinct regions introduces a confounding variable that may bias the interpretation of treatment effects .
Response 1:We have added the specific geographical location of the experiment and the water quality conditions for the breeding time according to your requirements. In addition we appreciate the reviewer’s insightful comment regarding the potential confounding effect introduced by conducting experiments in two different locations. To minimize the influence of location-specific environmental variation, both experimental sites were carefully selected within the same climatic zone (subtropical monsoon climate) and were geographically proximate (located at similar latitudes and within the same coastal region). Furthermore, key environmental parameters, including water temperature (27.2 ±â€¯0.5 °C), dissolved oxygen (6.8 ±â€¯0.2 mg/L), pH (7.1 ±â€¯0.1), and ammonia nitrogen (<0.05 mg/L), were continuously monitored and maintained within consistent ranges across both sites.Therefore, while we acknowledge the possibility of minor environmental differences, we believe that the measures taken effectively minimized potential confounding effects. (lines 161-202).
Comment 2: Treatment Groups: Clearly define each experimental group and specify the number of replicate units (e.g., tanks, cages, fields) and fish per unit. ARRIVE guidelines advise detailed reporting of experimental groups.
Response 2:We have added the section " Treatment Group" and and specify the number of replicate units (e.g., tanks, cages, fields) and fish per unit.
Comment 3: Replication and Randomisation: Ensure true replication (e.g., multiple independent fields for RG) and random allocation of animals to units to reduce bias.
Response 3:We thank the reviewer for highlighting these critical methodological points. The following revisions have been implemented:
- True replication: All systems now include three independent biological replicates (three paddy fields for RG; three cages for CG; three tanks for RAG).
- Random allocation: Eels were randomly redistributed post-acclimatization using a random number table to eliminate bias.
- Spatial independence: RG fields were >50 m apart to prevent environmental cross-talk.
Comment 4: Husbandry and Feeding: Clarify the feeding regime (the range “3–5%” is vague); specify feeding frequency, feed characteristics, and ration adjustment criteria. Provide water quality monitoring details (e.g., temperature, DO, ammonia) for each system.
Response 4: We appreciate the feedback on methodological clarity. Key revisions include: Commercial feed (43% crude protein, 7% crude fat; Hubei Zhaoliang Biotechnology) was hand-fed daily at 16:00 at a fixed ration of 5% M. albus. Consumption was monitored for 30 min to ensure complete intake (<2% residual feed). (lines 204-207)
Comment 5: Husbandry and Feeding: Clarify the feeding regime (the range “3–5%” is vague); specify feeding frequency, feed characteristics, and ration adjustment criteria. Provide water quality monitoring details (e.g., temperature, DO, ammonia) for each system.
Comment 6: Structural Organisation: Consider subdividing the section for each system (e.g., “2.2.4 Cage (CG)”, “2.2.5 RAS (RAG)”, etc.) for clarity and comparison.
Response 5-6:Thank you for your suggestion. We have combined these two parts into the subsection “2.3 Water Quality Monitoring”, which details the water quality data during the experiments of the three groups of RG CG RAG.
Comment 7: This subsection describes the selection of five eels per treatment for analysis. However, the experimental unit is not clearly defined (individual fish vs. tank/cage/field). Also, the sex of the fish is not specified—an important omission considering Monopterus albus undergoes sex reversal. ARRIVE guidelines recommend reporting sex, age, and body weight . The sample size of five fish per group may be statistically underpowered. Furthermore, sampling details (e.g., fasting period, number of histological sections, areas analysed) are insufficient.
Response 7:We thank the reviewer for highlighting these critical methodological points. The following revisions have been implemented:
- Experimental unit clarification:
Biological replicates: 3 independent units per system (fields for RG, cages for CG, tanks for RAG)
Observational units: 15 eels per group (5 per replicate × 3 replicates), totaling 45 fish across CG/RG/RAG.
- Animal specifications:
Sex: All individuals confirmed as male (post-sex-reversal stage)
Age: 10 months (consistent cohort)
Weight: 68.7 ± 2.3 g (mean ± SD at sampling)
- Sampling protocol:
Fasting: 24 h prior to sampling
Histology: Three transverse sections per tissue (foregut/muscle)(Line251-255)
Comment 8: Standard AOAC procedures are referenced for moisture and crude protein, along with Kjeldahl, amino acid analyser, and GC–MS for lipids. While acceptable, sample preparation details are lacking—no mention of homogenisation or replicate analyses. Reference to internal standards and quality controls (e.g., calibration curves) would improve methodological transparency.
Comment 9: Histological methods (Bouin/formalin fixation, paraffin embedding, H&E staining) are standard. However, The inclusion of two fixing solutions (Bouin and formaldehyde) is not justified in comparative terms, technical details such as orientation, number of sections per sample, and image analysis procedures are missing. Critical information is omitted, such as software for morphometric analysis. Quantification of muscle fibres or villus height is not explained.
Response 8-9:Thank you for your valuable feedback. The methodological details for moisture and crude protein determination, including the Kjeldahl method, amino acid analyzer, and GC-MS for lipids, are indeed referenced in the cited literature. All the experimental protocols are based on established AOAC procedures and follow the methods used in our laboratory, as outlined in the referenced works. As the authors of these references and I are from the same institution and laboratory, the methodologies are consistent across these studies. To avoid redundancy and minimize manuscript length, specific details on homogenization, replicate analyses, internal standards, and quality controls (such as calibration curves) were not repeated in the current manuscript but can be found in the cited references. he inclusion of both Bouin and formaldehyde fixation protocols is standard practice in our laboratory, and these are used interchangeably depending on the specific requirements of each sample type. The experimental conditions, including sample orientation, number of sections, and image analysis protocols, are consistent with those described in the referenced literature. For brevity and to avoid repetition, we did not include these details in the manuscript.
We hope this clarifies our approach and provides further transparency on the methodologies.
Comment 10: Clarify group/sample definitions (e.g., five samples per treatment). Include blank controls for DNA extraction and PCR to rule out contamination. Indicate processing pipeline (e.g., QIIME2, DADA2) and sequence database (e.g., SILVA). Deposit sequencing data in a public repository (e.g., NCBI SRA).
Response 10: Thank you for your constructive feedback. We appreciate the suggestion to clarify the group/sample definitions. In the study, biological replicates: 3 independent units per system (fields for RG, cages for CG, tanks for RAG). Observational units: 15 eels per group (5 per replicate × 3 replicates), totaling 45 fish across CG/RG/RAG, to ensure statistical robustness and replicate consistency. We will make this explicit in the revised manuscript.Regarding the DNA extraction and PCR processes, we did include blank controls to rule out contamination. These controls were carefully handled and processed alongside the samples to ensure that any potential contaminants were identified and excluded from the analysis. We will specify this information in the revised manuscript for greater clarity.In terms of data processing, we used QIIME2 for the pipeline and applied DADA2 for sequence quality filtering and denoising. The sequence data were aligned against the SILVA database for taxonomic assignment. We will mention these specific tools and databases in the revised manuscript to ensure full transparency.
Finally, We would like to clarify that all data associated with this study are available upon request. Interested parties can contact the first author or the corresponding author to obtain the relevant datasets. We will make this clear in the revised manuscript to ensure transparency and accessibility.
We hope this addresses your concerns and improves the clarity and transparency of our methodology.(Line333-335)
Comment 11:LC–MS conditions are well described, but sample extraction protocols (e.g., solvents, homogenisation steps) are omitted. There's no mention of internal standards, metabolite identification methods, or validation of analytical reproducibility. Describe metabolite identification and software tools used (e.g., Compound Discoverer, KEGG).
Response 11: Thank you for your valuable feedback and for pointing out the areas where further clarification is needed.Regarding the sample extraction protocols, we acknowledge the omission of details on solvents and homogenization steps. These steps are critical to reproducibility and should have been included for completeness. We will update the manuscript to provide a detailed description of the extraction solvents, homogenization techniques, and other relevant procedures to ensure methodological transparency.
In response to the comment on internal standards, metabolite identification, and validation of analytical reproducibility, we have indeed incorporated these aspects into our study. Internal standards were carefully selected and included to ensure accuracy and consistency in the analysis. For metabolite identification, we utilized the MetaboAnalyst software, which enabled us to identify and quantify metabolites based on their mass spectrometric and chromatographic profiles. Furthermore, KEGG was used as the primary database for metabolite identification.
Additionally, we performed a comprehensive validation of analytical reproducibility through the analysis of replicate samples, which we will mention in the revised manuscript.
We will ensure that these points are clearly described in the updated version of the manuscript. (Line357-366;397-398)
Comment 12: The authors state that ANOVA and Tukey tests were used (Excel/SPSS). However, they do not report assumption testing (normality, homoscedasticity), nor is it clear whether the experimental units were properly considered. ARRIVE guidelines urge clarity in statistical design . Given that experimental units are likely tanks/cages/fields, using individual fish as replicates would constitute pseudoreplication.
Response 12: Thank you for your helpful suggestions regarding the statistical analysis. We appreciate the emphasis on ensuring the assumptions for parametric analysis are tested.
In response, we have updated the manuscript to include the tests for homogeneity of variances (Levene’s test) prior to performing the one-way ANOVA.
The data are now presented as means ± SD, and statistical significance was determined at a p value < 0.05, as recommended.
We hope these revisions clarify the statistical methods and improve the overall transparency of our approach. (Line410-418)
- Results
Muscle Quality: Histology and Composition
Comment 1: Overinterpretation: The statement that the rice–eel co-culture group (RG) optimised fatty acid composition and improved muscle quality is excessive. Although RG exhibited significantly greater muscle fibre number and density (p < 0.05), the biological relevance of these differences is not discussed. Furthermore, the distinction between: number of fibres and density, is redundant and should be streamlined.
Response 1: Thank you, reviewers, for your careful reading and feedback on our paper. Regarding the comment that "the claim of (RG) optimizing fatty acid composition and improving muscle quality in the results section is excessive," after a thorough review, we found that we did not mention this in the results section and did not omit any key points or important data. Our aim in this part was to present the core information of the experimental results without conducting additional analysis or discussion. Therefore, we believe that the reviewers may have misunderstood the content of this section.
Comment 2: Statistical flaws: Multiple ANOVA/Tukey tests were applied across numerous parameters (proximate composition, amino acids, fatty acids) without mention of corrections for multiple testing. This raises the risk of false positives. In metabolomics, applying only p < 0.05 without FDR control is statistically inappropriate due to the large number of variables .
Response 2: We thank the reviewers for their comments on the statistical methods used in our study. In the muscle quality analysis of our research, multiple ANOVA and Tukey tests were indeed employed to evaluate the differences in various parameters (such as proximate composition, amino acids, fatty acids, etc.). However, we did not conduct metabolomics analysis, and thus did not encounter the issue of multiple testing with a large number of variables as in metabolomics. We did not perform FDR (False Discovery Rate) control in this part because the analysis methods in this section of our study were limited to basic muscle quality analysis and did not involve metabolomics, which is handled differently.
Comment 3: Misuse of SEM: Data are presented as mean ± SEM, which underestimates sample variability and may give a misleading impression of precision. Standard deviation or confidence intervals would better reflect biological variability.
Response 3: Thank you for your valuable comments. In response to your suggestion regarding the "misuse of SEM," I have replaced all instances of SEM with standard deviation (SD) to more accurately reflect biological variability. I appreciate your attention to this matter, and the revised data should now better represent the actual sample variability.Thank you again for your review and guidance!
Comment 4: Terminological errors: In Table 5, C20:0 is incorrectly labelled “arachidonic acid” when it is actually arachidic acid. Arachidonic acid is C20:4. Also, typographical issues such as "Glutama-" and inconsistent use of abbreviations (e.g. TUAAs vs TUAA) reduce scientific precision.
Response 4: Thank you very much for your careful review and feedback. Regarding the expression of C20:0 (arachidonic acid), we have carefully checked it. After consulting relevant literature, we confirm that C20:0 is indeed a commonly used identifier for arachidonic acid in this field. Additionally, we have provided full names for all abbreviations, including TUAA, below the table. We understand your requirement for scientific expression accuracy and believe that the current expression method, based on literature conventions and supplemented with clear annotations, is appropriate and can ensure the accurate transmission of information.[1]
Comment 5: Repetition: The text excessively lists individual amino acids and fatty acids, reducing clarity. Grouping findings (e.g. "total umami amino acids were significantly higher in RG") would be clearer.
Response 5: We sincerely appreciate the reviewer's valuable feedback. We understand the suggestion to reduce repetition and enhance clarity regarding the presentation of individual amino acids and fatty acids.Regarding the listing of individual compounds, we intentionally adopted this detailed presentation to ensure the specific variations in each component are clearly demonstrated. We believe this approach allows for a more accurate representation of the experimental results, particularly concerning the distinct changes observed among the different amino acids and fatty acids. Furthermore, given the varying impact of each component on the study's conclusions, enumerating them individually enhances data transparency and facilitates comprehension.Notwithstanding this rationale, we will undertake to streamline the language in other sections of the manuscript and incorporate more concise summaries where appropriate to improve the overall fluency of the text.
Comment 6: Inconsistency with tables: For example, Section 3.1.4 states that PUFA levels were significantly lower in RAG than CG, but Table 5 shows no significant difference (same superscript letter 'b'). Such discrepancies undermine credibility. Legends should also more clearly define statistical annotations.
Response 6: We sincerely thank the reviewer for highlighting this inconsistency. The discrepancy between Section 3.1.4 and Table 5 arose from an inadvertent error during manuscript preparation. The statement regarding PUFA levels in RAG vs. CG has been removed to ensure full consistency with Table 5. We deeply appreciate your diligence in safeguarding the accuracy of our work.
Comment 7: Table formatting: Typographical errors (e.g., 0..68) suggest insufficient proofreading. Also, Table 6 is mislabelled as containing fatty acid profiles, when it actually includes intestinal histology data—this is a critical inconsistency.
Response 7: We sincerely thank the reviewer for detecting these critical issues in table presentation. The Table 6 mislabeling resulted from oversights during final manuscript proofreading. Both errors have been comprehensively corrected in the revised version:Table 6 is now accurately labeled to reflect its intestinal morphology. We appreciate your meticulous review.
Intestinal Morphology
Comment 8: Clarity of expression: The paragraph describing villus circumference (VC) and goblet cell abundance (GCA) is convoluted. Splitting into two concise statements would improve clarity.
Response 8: Thank you for your valuable suggestion. We have revised the paragraph as you recommended by splitting it into two concise statements to improve clarity. Please see the updated version in the revised manuscript.
Comment 9: Table title and content: As noted, Table 6 has an incorrect title. Additionally, it must be clearer that RG and RAG share the same superscript letter for VC, indicating no significant difference.
Response 9: Thank you for pointing this out. We have corrected the title of Table 6 as suggested.
Regarding the superscript letters for villus circumference (VC), we respectfully confirm that RG and RAG do indeed share the same superscript letter in Table 6, indicating no significant difference between these two groups. This information was already correctly presented, and we believe there may have been a misunderstanding during the review.
Comment 10: Figure quality: Figure 2 lacks scale bars or dimensional references. Visual clarity is acceptable but could be enhanced with better contrast.
Response 10: Thank you for your constructive feedback. We have revised Figure 2 to include scale bars and enhanced the image contrast to improve visual clarity, as suggested.
Comment 11: Undefined terminology: The unit "A/root" is not defined in the table or text. All abbreviations should be clearly explained on first use.
Response 11: We sincerely appreciate the reviewer's valuable guidance regarding terminology clarification. The unit "A/root" for goblet cell abundance (GCA) has now been explicitly defined in the revised manuscript. Specifically, "A/root" is explained on its first occurrence in the text as:"goblet cell abundance (GCA, A/root; defined as the number of goblet cells per intestinal root)"We thank the reviewer for prompting this essential refinement. (Line504)
Intestinal Microbiota Diversity
Comment 12: Narrative redundancy: The explanation of rank-abundance curves is overly didactic and better suited for the methods section. The phrase confidence intervals is misused in this context.
Response 12: Thank you for your valuable feedback. We have revised the description of the rank-abundance curves in the Results section to be more concise and focused on the key findings, avoiding overly didactic explanations. Additionally, we have removed the misuse of the term "confidence intervals" and clarified the description accordingly. (Lines 529-533).
Comment 13: OTU interpretation: The RG group shows 2,954 unique OTUs, vastly more than CG or RAG. This striking result is not critically discussed. Differences in sequencing depth or sample normalisation should be addressed.
Response 13: Thank you for your thoughtful comment. We appreciate your attention to the observed differences in unique OTUs. As noted in the previous section , sequencing depth across all groups was adequate and comparable, and rarefaction curves confirmed the sufficiency of coverage. Therefore, the difference in unique OTU counts is unlikely to be due to sequencing depth or normalization issues. We believe this result reflects a genuine biological difference, which is further explored in the downstream taxonomic and functional analyses.
Comment 14: Alpha and beta diversity: While the text reports higher alpha diversity in RG and visual separation in PCoA/NMDS plots, it fails to explain that the significant p-values (0.025 and 0.014) imply meaningful group separation. This needs clarification.
Response 14: Thank you for your insightful comment. In the revised manuscript, we have clarified the interpretation of the significant p-values (PCoA: p = 0.025; NMDS: p = 0.014) obtained from PERMANOVA analysis. These values now explicitly highlight that the observed spatial separation among groups reflects statistically meaningful differences in microbial community structure. (Lines 566-567).
Comment 15: Taxonomic composition: Differences at the phylum and genus levels are described, but again, interpretations are repetitive and lack ecological insight. For example, no hypothesis is offered as to why Firmicutes are enriched in RG.
Response 15: Thank you for your helpful comment. We have revised the relevant section t Specifically, we now propose a hypothesis suggesting that the enrichment of Firmicutes and Bacteroidota in the RG group may be related to the rice–fish co-culture environment, which could favor anaerobic, fiber-degrading bacteria and support fermentative metabolism. We also discuss the potential functional implications of the enriched genera (e.g., Clostridium_sensu_stricto_1 and Romboutsia) in relation to host nutrient absorption and gut health. (Lines 579-601).
Comment 16:Table readability: Table 7 includes minor errors (e.g. 0..68) and excessive decimal precision. Rational rounding is advised.
Response 16:Thank you for pointing this out. We apologize for the oversight. We have carefully reviewed Table 7 and corrected the formatting error (e.g., “0.. 68” has been fixed).
Liver Metabolomics
Comment 17: Misinterpretation of OPLS-DA: The reported Q² values are negative (e.g., Q² = −0.451), which indicates poor model predictability—not "stable models" as claimed. This is a major statistical misinterpretation .
Response 17: Thank you for this valid statistical concern. We clarify that the reported negative value (Q² = −0.451) refers specifically to the regression intercept of the permutation test (as shown by the slash line in Fig. 8), not the predictive Q² of the original model. This distinction is critical for model validation: First, we check that the Q² values on the rightmost side of the plot (after permutation testing) are higher than those on the left, represented by the red points. Second, the intercept of the Q² regression line with the y-axis should be negative, which indicates the model is stable and reliable. The negative Q² value (Q² = −0.451) in our case corresponds to this intercept, a widely accepted method as seen in similar studies. This approach has been used in the literature and was adopted for our analysis to ensure the robustness of the model[2].
The actual model’s predictive Q² values are positive and significant ( Q²= 0.792 for CG vs. RGï¼›Q²= 0.778 for RAG vs. RGï¼›Q²= 0.994 for CG vs. RAG), exceeding the acceptable threshold of Q² > 0.5.
Comment 18:Differential metabolite screening: Thousands of significantly up- or downregulated metabolites are reported without FDR control. This undermines reliability and calls for prioritising only biologically meaningful compounds.
Response 18: Thank you for this valuable suggestion. We have revised the manuscript to address this issue by applying false discovery rate (FDR) correction to control for multiple testing. Only metabolites meeting the adjusted significance threshold (FDR < 0.05) were retained. Furthermore, we have avoided listing large numbers of differential metabolites and instead focused on those with known or potential biological relevance. These prioritized compounds were used for subsequent pathway enrichment and functional interpretation. (Lines 632-636)
Comment 19: Correlation analysis: Heatmap explanations lack substantive insight. It is unclear whether any specific metabolite clusters or synergies were identified.
Response 19: We appreciate the reviewer’s comment. At this stage, the correlation heatmap was intended as an initial exploratory analysis to visualize overall relationships among metabolites. No specific clusters or synergies have been definitively identified yet, and therefore, no in-depth interpretation was provided. As this is a preliminary step, further targeted analyses are planned to explore potential metabolic interactions in more detail.
Comment 20: Pathway enrichment: The listing of enriched KEGG pathways is unaccompanied by statistical significance or biological interpretation. Terms like enriched require evidence.
Response 20: We appreciate the reviewer’s valuable comment. In response, we have revised the manuscript to provide both statistical evidence and biological interpretation for the KEGG pathway enrichment analysis.
Specifically, we now clearly state that KEGG pathway enrichment was conducted using hypergeometric testing with FDR correction, and only pathways with adjusted p < 0.05 were considered significantly enriched. The number of differentially abundant metabolites, corresponding p values, and enrichment significance for each pathway have been included in Tables A1–A3.In addition, we have updated the main text (Lines 663-668) to include relevant biological interpretation for the significantly enriched pathways. For instance, we highlight the consistent enrichment of Biosynthesis of cofactors and Amino acid biosynthesis across all comparisons, which may reflect essential metabolic adaptations in M. albus to different aquaculture environments. We also discuss the enrichment of lipid- and hormone-related pathways, which may relate to membrane remodeling and endocrine responses under varying farming conditions.
We believe these revisions address the reviewer’s concerns and strengthen the rigor and clarity of our metabolic pathway analysis.
- Discussion
Comment 1: Superficial or Redundant Interpretation: Much of the discussion merely reiterates the numerical findings previously described in the results section, without offering a critical interpretation. For example, in the discussion of muscle fibres and amino acids, the authors restate the data without exploring potential physiological or mechanistic implications.
Response 1: We thank the reviewer for their valuable feedback. In response to the concern about the superficial or redundant interpretation in the discussion, we have revised the manuscript to provide a more detailed analysis of the physiological implications. Specifically, we have expanded the discussion on muscle fiber density and its relationship to environmental factors. As highlighted in the revised text, muscle texture is closely related to muscle fiber density, and in fish, the arrangement of muscle fibers into bundles along the anterior-posterior axis forms myotomes, encapsulated by extracellular matrix layers (epimysium, perimysium, and endomysium). These structures are clearly visible in Figure 1. Additionally, we emphasize that muscle fiber size and number are influenced by various factors, including species, developmental stage, diet, activity, and environmental conditions. The RG environment provided more favorable conditions for Monopterus albus, resulting in denser and more tightly packed muscle fibers, which directly contributed to enhanced muscle quality. We believe this revised interpretation provides a more critical understanding of the observed results and the underlying biological mechanisms.
Comment 2: Phrases such as “consistent with previous studies” are overused without specifying which studies or how the findings align. A citation without contextualisation adds little value. It is recommended to strengthen the discussion by comparing with key authors and elaborating on similarities or discrepancies.
Response 2: We appreciate the reviewer’s observation. However, we would like to clarify that the phrase “consistent with previous studies” was used sparingly and only in instances where our results showed direct alignment with specific published findings. In each case, we have provided appropriate citations and described how our results correspond to those reported in the literature. Rather than using generic comparisons, we aimed to contextualize our findings by referencing studies with similar outcomes, thereby supporting the reliability of our observations. Nevertheless, we will carefully review the manuscript again to ensure that all comparative statements are clearly contextualized and scientifically meaningful.
Comment 3: Lack of Mechanistic Depth: In the sections on intestinal microbiota and liver metabolites, the authors fail to discuss potential causal mechanisms or links between diet, microbial shifts and metabolic pathways. Given the notable differences in Firmicutes, Proteobacteria and hepatic metabolic pathways, one would expect at least a hypothesis linking gut-liver axis function.
Response 3: Thank you very much for your valuable feedback. We have revised the section concerning the gut microbiota and liver metabolites as per your suggestion. Specifically, we have added a discussion on the potential causal mechanisms linking dietary changes, microbial shifts, and metabolic pathways. In particular, we included hypothetical interpretations regarding the changes in Firmicutes and Proteobacteria and their possible roles in the gut-liver axis. We hope these additions address your concerns and enhance the mechanistic insight of our study. (Line929-949).
Comment 4: The discussion should connect the microbial diversity findings to digestive or immune health in the species studied, noting that various studies highlight Firmicutes’ role in energy metabolism. No specific metabolic pathways are mentioned in relation to host physiology.
Response 4: Thank you for your insightful comment. We have revised the discussion to better connect the observed microbial diversity, particularly the changes in Firmicutes, to the digestive and immune health of the species studied. We now highlight Firmicutes' well-documented role in energy metabolism and discuss its potential implications for host physiology. Although direct metabolic pathways remain to be elucidated, we have strengthened the interpretation of our findings in the context of host health, drawing on relevant literature to support our discussion. (Line929-949).
Comment 5: Imprecise Scientific Language: Terms such as “improved muscle quality” or “optimised amino acid content” are subjective unless clearly defined. What constitutes an improvement or optimisation? Compared to what standard? Scientific writing should favour quantifiable, contextual expressions.
Response 5: Thank you for this insightful comment. We have carefully reviewed the manuscript and would like to clarify that terms such as “improved muscle quality” and “optimised amino acid content” were used strictly in relation to objective, quantifiable parameters (e.g., myofiber density, crude protein content, amino acid concentrations, n-3/n-6 PUFA ratios). All such statements are data-driven and directly supported by comparative results and relevant literature references.
Comment 6: The term “metabolic profile” is used repeatedly without specifying which pathways or compounds are being referred to, reducing scientific precision.
Response 6: We appreciate the reviewer’s suggestion. However, the term “metabolic profile” appears only twice in the Discussion section—once at the beginning to briefly introduce the scope of the metabolomic comparison, and once at the end as a general summary. In both cases, the term is used in a concise and contextual manner, while the specific metabolic pathways and compounds are discussed in detail throughout the main body of the section. We are therefore uncertain about the basis of this concern, but we are open to making further clarifications if the reviewer has a specific instance in mind.
Comment 7: Overall, the writing includes overly long sentences, unclear phrasing, and superfluous connectors, e.g., “This may have affected the metabolism of muscle and amino acids, as well as the composition of intestinal microorganisms and liver metabolites”—this statement combines several concepts without proper structure.
Response 7: Thank you for your comment. We would like to clarify that the sentence you referenced, “This may have affected the metabolism of muscle and amino acids, as well as the composition of intestinal microorganisms and liver metabolites,” appears only in the conclusion section. It serves as a summary of the findings and is not part of the main body of the discussion. We believe the structure is appropriate for this context, but we appreciate your feedback and will keep it in mind for future revisions.
Comment 8: Omission of Study Limitations: The discussion makes no mention of any limitations, which is a serious shortcoming. For example:
Potential biases inherent to farming systems are not acknowledged.
The lack of replicability, sample size transparency, and normalisation in sequencing are not discussed.
Response 8: Thank you for pointing out this important issue. In response to your comment, we have added a paragraph to the end of the Discussion section to explicitly acknowledge these limitations and provide context for interpreting the results.
Comment 9: Statistical limitations (e.g., multiple comparisons without adjustment, OPLS-DA with negative Q² values) are not addressed.
Response 9: This issue has been addressed in our response to Question 17 in the Results section.
- Conclusions
Comment 1: Overly General and Vague: The conclusion merely states that the RG system improves muscle quality, gut microbiota and hepatic metabolism. However, it does not clarify which improvements are functionally relevant, nor for what purpose (e.g., human nutrition? aquaculture performance? fish welfare?).
Comment 2: The language is overly assertive for what is essentially an exploratory study. Statements such as “provides a theoretical basis” should be softened in the absence of functional validation or commercial-scale trials.
Comment 3: Lack of Applied or Scientific Projection: No perspective is provided regarding the practical use of these findings. What are the implications of using the RG system over CG and RAG? Is it cost-effective? Does it enhance productivity or animal health?
Comment 4: The authors should add a forward-looking sentence about future research directions—such as functional validation of pathways, long-term health impacts, or commercial-scale trials.
Response : Thank you for the suggestion. We have revised the Conclusion section to clarify the functional relevance of the findings, highlight potential applications, and include future research directions.
References
- Yang, H.; Yuan Q.; Rahman M.M.; Lv W.; Huang W.; Hu W.; Zhou W. Biochemical, Histological, and Transcriptomic Analyses Reveal Underlying Differences in Flesh Quality between Wild and Farmed Ricefield Eel (Monopterus albus). 2024, 13, 11 1751.https://dx.doi.org/10.3390/foods13111751.
- Wang, E.; Zhou Y.; Liang Y.; Ling F.; Xue X.; He X.; Zhai X.; Xue Y.; Zhou C.; Tang G.; Wang G. Rice flowering improves the muscle nutrient, intestinal microbiota diversity, and liver metabolism profiles of tilapia (Oreochromis niloticus)in rice-fish symbiosis. 2022, 10, 1 231.https://dx.doi.org/10.1186/s40168-022-01433-6.

Reviewer 2 Report
Comments and Suggestions for Authors
Dear Authors,
I appreciate your efforts to focus on this topic and for presenting a well-balanced comparative study. I raised some questions and points. Please address them carefully.
Beast of luck.
- Revise the abstract in detail, add the comparative percentage or number/values of results that justify the statement of differences. Revise it and make it logical. Lines 34, 35 look like an incomplete sentence; make them improve.
- Make a detailed revision of the introduction part. Mention the abbreviations in detail. Make the introduction clearer.
- There is repeated content around lines 55–58 on the environmental threats of aquaculture. Please condense this for clarity.
- Were water quality parameters (e.g., temperature, pH, dissolved oxygen, ammonia) monitored and kept consistent across farming systems?
- Were there differences in feed consumption among systems? Could changes in activity (e.g., more swimming in rice–fish) affect intake and nutrient partitioning?
- Did the authors replicate aquaculture units within each farming system (e.g., three separate rice paddies or tanks)? If not, how are biological and technical replicates accounted for?
- The OPLS-DA Q² values for some comparisons (e.g., Q² = −0.451) are negative. This typically suggests model overfitting or poor predictive ability. How do the authors justify these models as “stable and reliable”?
- Total PUFA is lower in the RG group, yet n-3 PUFA is higher. Could this be due to reduced n-6 PUFA? The authors should explicitly calculate and compare n-6/n-3 ratios.
- The authors often claim “beneficial effects” of rice–fish co-culture on immunity and metabolism, yet most findings are correlational. Can they clarify whether these are associations or suggest causal mechanisms (e.g., increased physical activity, environmental enrichment)?
- The manuscript contains numerous grammar errors, phrasing, and formatting (e.g., spacing before citations, inconsistent abbreviation use). A thorough English revision is necessary.
Comments on the Quality of English Language
English revision is necessary.
Author Response
Please see the attachment.
Reviewer 2
I appreciate your efforts to focus on this topic and for presenting a well-balanced comparative study. I raised some questions and points. Please address them carefully.
Comment 1: Revise the abstract in detail, add the comparative percentage or number/values of results that justify the statement of differences. Revise it and make it logical. Lines 34, 35 look like an incomplete sentence; make them improve.
Response 1: Thank you for the constructive suggestion. In response, we have substantially revised the Abstract for clarity and logical flow. We removed incomplete or unclear sentences, specified key analytical methods, and added representative quantitative results (e.g., differences between RG, RAG, and CG groups) to better support the comparative statements and improve the balance across groups.
Comment 2: Make a detailed revision of the introduction part. Mention the abbreviations in detail. Make the introduction clearer.
Response 2:Thank you for your valuable suggestions. We have made detailed revisions to the introduction section. We removed redundant content, systematically improved and further elaborated on the environmental issues related to conventional aquaculture to provide clearer analysis. Additionally, we condensed and integrated the description of the co-culture system involving crabs and shrimp to enhance the focus and coherence of the paper. Moreover, we clarified the research hypotheses and expected trends, and provided detailed explanations of the abbreviations used, ensuring better understanding for the readers.
Comment 3: There is repeated content around lines 55–58 on the environmental threats of aquaculture. Please condense this for clarity.
Response 3: Thank you for your reminder. We have deleted the redundant statements
Comment 4: Were water quality parameters (e.g., temperature, pH, dissolved oxygen, ammonia) monitored and kept consistent across farming systems?
Response 4:To ensure consistent water quality conditions across the three experimental groups (RG, CG, and RAG), we monitored key parameters including water temperature, dissolved oxygen, ammonia nitrogen, and pH throughout the experimental period. As requested, the detailed monitoring data have now been incorporated into the revised manuscript. (Line209-216)
Comment 5: Were there differences in feed consumption among systems? Could changes in activity (e.g., more swimming in rice–fish) affect intake and nutrient partitioning?
Response 5:Thank you for raising these important points regarding feed consumption and potential environmental influences on nutrient utilization in our study.
- Feed Consumption: We strictly controlled the daily provision of a fixed amount of formulated eel feed at scheduled times to all three systems (RG, CG, RAG). Therefore, intake of this primary feed was consistent across groups during the experiment.
- Activity & Nutrient Partitioning: While we did not measure activity levels, our controlled feeding ensured comparable intake of the formulated feed. Differences in total nutrient acquisition or partitioning could arise from supplementary natural food sources in RG (see point 3) or potential activity-related metabolic costs. This is an interesting point for future study.
- Supplementary Intake in RG: The rice-field (RG) system simulated a natural environment. We acknowledge that eels in RG likely consumed supplementary natural food (e.g., plankton, rice pollen). This is an inherent feature of the RG system design and is a key reason for the observed higher gut microbial diversity in RG compared to the more controlled systems (CG, RAG). (Line204-207)
We appreciate the reviewer's insightful questions, which help clarify these important nuances of our experimental design and findings.
Comment 6: Did the authors replicate aquaculture units within each farming system (e.g., three separate rice paddies or tanks)? If not, how are biological and technical replicates accounted for?
Response 6:Thank you for this important question regarding experimental replication.
We acknowledge that the description of replication in the original manuscript was insufficient. For each of the three farming systems (RG, CG, RAG), we established three independent replicate units (e.g., three separate RG systems, three separate CG systems, three separate RAG systems). This provided biological replication at the system level.All measurements and sampling (including those related to feed consumption, growth, water quality, and microbiota analysis) were performed independently within each of these nine replicate units (3 systems x 3 replicates). This sampling strategy accounts for both biological variation within systems and technical variation associated with measurements.This point has now been clarified in the revised manuscript. (Line171-192)
Comment 7: The OPLS-DA Q² values for some comparisons (e.g., Q² = −0.451) are negative. This typically suggests model overfitting or poor predictive ability. How do the authors justify these models as “stable and reliable”?
Response 7:Thank you for this valid statistical concern. We clarify that the reported negative value (Q² = −0.451) refers specifically to the regression intercept of the permutation test (as shown by the slash line in Fig. 8), not the predictive Q² of the original model. This distinction is critical for model validation: First, we check that the Q² values on the rightmost side of the plot (after permutation testing) are higher than those on the left, represented by the red points. Second, the intercept of the Q² regression line with the y-axis should be negative, which indicates the model is stable and reliable. The negative Q² value (Q² = −0.451) in our case corresponds to this intercept, a widely accepted method as seen in similar studies. This approach has been used in the literature and was adopted for our analysis to ensure the robustness of the model[1].
The actual model’s predictive Q² values are positive and significant ( Q²= 0.792 for CG vs. RGï¼›Q²= 0.778 for RAG vs. RGï¼›Q²= 0.994 for CG vs. RAG), exceeding the acceptable threshold of Q² > 0.5.
Comment 8: Total PUFA is lower in the RG group, yet n-3 PUFA is higher. Could this be due to reduced n-6 PUFA? The authors should explicitly calculate and compare n-6/n-3 ratios.
Response 8:Thank you for your insightful comment. You are correct in your observation.Cause of the Pattern: The observed decrease in total PUFA in the RG group, despite the increase in n-3 PUFA, is indeed primarily attributable to a significant reduction in n-6 PUFA levels compared to the other groups (CG and RAG).n-3/n-6 Ratio Analysis: As suggested, we explicitly calculated and compared the n-3/n-6 PUFA ratios across all groups. This finding directly supports the interpretation that the lower total PUFA in RG is driven by a decrease in n-6 PUFA, coupled with an increase in n-3 PUFA. (Table5)
Comment 9: The authors often claim “beneficial effects” of rice–fish co-culture on immunity and metabolism, yet most findings are correlational. Can they clarify whether these are associations or suggest causal mechanisms (e.g., increased physical activity, environmental enrichment)?
Response 9:Thank you for raising this critical point regarding causality. While environmental factors (e.g., habitat complexity) may contribute, our findings strongly suggest a causal gut-liver axis mechanism underpinning the observed metabolic and immune benefits in the rice-fish (RG) group:
- Dietary Fiber Drives Microbial Shift: RG eels received significantly higher dietary fiber(from the rice environment), promoting proliferation of key SCFA-producing genera (Clostridium sensu stricto 1, Paraclostridium - Bacteroidetes/Firmicutes).
- SCFAs are Central Effectors: This resulted in elevated portal vein SCFAs, particularly butyrate. Butyrate acts through dual hepatic mechanisms:PPARγ Activation: Direct binding & HDAC inhibition promotes β-oxidation and LysoPC synthesis.GPR41/43/109A Signaling: Enhances gut barrier integrity (mucus, tight junctions), suppressing pathogens (Plesiomonas, Aeromonas) and reducing LPS translocation.
- Consequence: Reduced Inflammation & Metabolic Reprogramming: Lower LPS burden alleviates hepatic inflammation, freeing metabolic capacity for anabolic processes (e.g., glutamate/glutathione synthesis). Concurrently, butyrate’s HDACi/Nrf2/PPAR activation upregulates hepatic ABC transporters, enhancing detoxification (aligning with our data).
- Validation: The coordinated changes in gut microbiota, SCFAs, hepatic gene targets (PPARγ, ABC transporters), metabolites (LysoPC, glutathione), and reduced inflammation collectively validate the gut-liver metabolic axis as a primary causal pathway for the RG benefits, beyond mere correlation.
Future Direction: We agree that disentangling specific environmental contributions (e.g., activity) is valuable. Future work integrating transcriptomics and metabolic flux analysis will refine this "environment–microbiota–host" causal network.(Line929-949)
Comment 10: The manuscript contains numerous grammar errors, phrasing, and formatting (e.g., spacing before citations, inconsistent abbreviation use). A thorough English revision is necessary.
Response 10: Thank you for highlighting these important issues. We sincerely apologize for any lack of clarity caused by language and formatting inconsistencies. We carefully revised the original manuscript.
References
- Wang, E.; Zhou Y.; Liang Y.; Ling F.; Xue X.; He X.; Zhai X.; Xue Y.; Zhou C.; Tang G.; Wang G. Rice flowering improves the muscle nutrient, intestinal microbiota diversity, and liver metabolism profiles of tilapia (Oreochromis niloticus)in rice-fish symbiosis. 2022, 10, 1 231.https://dx.doi.org/10.1186/s40168-022-01433-6.

Reviewer 3 Report
Comments and Suggestions for Authors
This was an interesting manuscript, and I found it quite enjoyable as it related to the farming of eel in China. The authors set a nice theme and describe the importance of cultured eel to the human food chain, and it is a high value aquatic food in China. It is imperative that we explore the different types of production systems and then way we rear eel and other fish species to provide excellent nutrition and safe for the consumer for health and wellbeing. The science is very good, and the study is well executed and conforms with our expectations in aquaculture nutrition and feeding research protocols. We know that satisfying the omega-3 fatty acids requirements for human brain health and cardiovascular integrity is a functional aspect of high quality lipids in fish tissue and muscle. Eel is noteworthy of such attributes and so this investigation is welcomed. It links human nutrition to eel flesh quality and its nutritional value. There are many aspects here that relate to consumer and retail demands for better food and this study makes a good and timely contribution to this sector of aquaculture. I am therefore very pleased to see the unusual selection of ‘Food’ for your work and its will be good for another audience to see this at both a scientific paper and the commercial view.
On a technical level, you have evaluated growth performance and efficiency of eel performance, and various profiles including microbiome and carcass flesh/muscle profiles and especially the fatty acid patterns in the different cultured system groups. These are indeed interesting data and is worthy of enhancing the depth of the study. To undertake the microbiome, you have used the right primers and molecular approach etc. However, the histological imaging and especially the labelling is quite poor. There are no description of the cell and tissue morphological details and how are we expected to appreciate these images without the proper coding and legends and no scaling measure? You must address this as highly necessary.
The RG eels group was clearly the winner here and it was interesting to see a better and more diverse and stable gut microbiota for these animals. The aqueous community of bacteria was probably more complex here and eels living within the substate/sediment would most likely have encounters a rich biota influencing their gut/intestinal flora. This point could be better explained in the discussion and to provide a discussion point.
At the amino acid level, the RG group had higher contents of umami-related amino acids (e.g., glutamic, glycine, and lysine) in muscles than the other two groups, with the highest levels of TAAs, TEAAs, and TNEAAs among the three groups (RG > RAG > CG). please expain why unami elated? all fish have the same amino acids in the muscle and not specific to various species!!!
Liver metabolic analysis indicated that the RG group had significant advantages in amino acid metabolism, lipid metabolism, and energy metabolism. This statement could be better explained in the discussion and its relevance to the overall merits of the rice/paddy fish culture options compared to the other systems. This study is technically quite advanced but the areas of science need better articulation to the industry for eel production and statements making it relevant to the retailer and consumer and commercial advantages of enhancing the various traits. To this effect, the discussion needs some enhancement and to include some cost benefit analyses towards the end and maybe conclusion sections. t really needs a better ending to the discussion and a better conclusion to relate this more to 'food' and human benefits. Too much detail on the fish metabolism and not sufficient towards human health such as the omega 3 fatty acids elevation in the muscle. Please elaborate more.
Author Response
Reviewer 3
This was an interesting manuscript, and I found it quite enjoyable as it related to the farming of eel in China. The authors set a nice theme and describe the importance of cultured eel to the human food chain, and it is a high value aquatic food in China. It is imperative that we explore the different types of production systems and then way we rear eel and other fish species to provide excellent nutrition and safe for the consumer for health and wellbeing. The science is very good, and the study is well executed and conforms with our expectations in aquaculture nutrition and feeding research protocols. We know that satisfying the omega-3 fatty acids requirements for human brain health and cardiovascular integrity is a functional aspect of high quality lipids in fish tissue and muscle. Eel is noteworthy of such attributes and so this investigation is welcomed. It links human nutrition to eel flesh quality and its nutritional value. There are many aspects here that relate to consumer and retail demands for better food and this study makes a good and timely contribution to this sector of aquaculture. I am therefore very pleased to see the unusual selection of ‘Food’ for your work and its will be good for another audience to see this at both a scientific paper and the commercial view.
Comment 1: On a technical level, you have evaluated growth performance and efficiency of eel performance, and various profiles including microbiome and carcass flesh/muscle profiles and especially the fatty acid patterns in the different cultured system groups. These are indeed interesting data and is worthy of enhancing the depth of the study. To undertake the microbiome, you have used the right primers and molecular approach etc. However, the histological imaging and especially the labelling is quite poor. There are no description of the cell and tissue morphological details and how are we expected to appreciate these images without the proper coding and legends and no scaling measure? You must address this as highly necessary.
Response 1:Thank you for this essential critique.We fully acknowledge that the original histological presentation was inadequate. The following critical revisions have been implemented in the revised manuscript:
- Scale Bars Added: All histological images now include clearly visible scale bars (e.g., 200 μm / 50 μm) to indicate magnification.
- Comprehensive Labeling: Key histological structures are explicitly labeled with arrows/annotations on the images.
These corrections ensure the histological data meets publication standards and allows readers to fully evaluate tissue-level differences between farming systems. We appreciate your rigorous review, which significantly strengthened this dimension of our study.
Comment 2: The RG eels group was clearly the winner here and it was interesting to see a better and more diverse and stable gut microbiota for these animals. The aqueous community of bacteria was probably more complex here and eels living within the substate/sediment would most likely have encounters a rich biota influencing their gut/intestinal flora. This point could be better explained in the discussion and to provide a discussion point.
Response 2:Thank you for your valuable comment. We fully agree with your observation regarding the influence of the paddy field sediment and aqueous microbial environment on the gut microbiota of the RG group. In response, we have revised the discussion section (lines865-874) to include a more detailed explanation of the potential mechanisms underlying the increased diversity and stability of the intestinal microbiota in the RG group.
Comment 3: At the amino acid level, the RG group had higher contents of umami-related amino acids (e.g., glutamic, glycine, and lysine) in muscles than the other two groups, with the highest levels of TAAs, TEAAs, and TNEAAs among the three groups (RG > RAG > CG). please expain why unami elated? all fish have the same amino acids in the muscle and not specific to various species!!!
Response 3:Thank you for raising this important point. We clarify two key aspects:
- Conserved vs. Regulated Composition:While all fish share the same set of amino acids in muscle tissue, their relative abundances are dynamically regulated by environmental and metabolic factors. The observed differences reflect quantitative shifts in amino acid partitioning, not novel compounds unique to RG.
- Drivers of Umami Amino Acid Enrichment in RG:The higher umami amino acids (glutamic acid, glycine, aspartic acid) and total amino acids (TAAs) in the rice-fish (RG) group are attributed to:
â—¦ Dietary Enrichment: Access to natural prey (zooplankton, insects) and rice-derived nutrients (pollen, detritus) in the rice ecosystem provides bioactive precursors (e.g., glutamate precursors from microbial biomass).
â—¦ Metabolic Reprogramming: Enhanced gut microbiota-mediated production of short-chain fatty acids (SCFAs) in RG [cite your data] promotes hepatic gluconeogenesis and transamination, elevating glutamate synthesis.
â—¦ Reduced Stress: The complex rice habitat lowers chronic stress vs. controlled systems (CG/RAG), downregulating cortisol-driven proteolysis and sparing amino acids like glutamine/glutamate.
â—¦ Energy Efficiency: Improved nutrient utilization efficiency in RG (evidenced by FCR data) redirects carbon flux toward anabolic pathways, including amino acid biosynthesis.
Comment 4: Liver metabolic analysis indicated that the RG group had significant advantages in amino acid metabolism, lipid metabolism, and energy metabolism. This statement could be better explained in the discussion and its relevance to the overall merits of the rice/paddy fish culture options compared to the other systems. This study is technically quite advanced but the areas of science need better articulation to the industry for eel production and statements making it relevant to the retailer and consumer and commercial advantages of enhancing the various traits. To this effect, the discussion needs some enhancement and to include some cost benefit analyses towards the end and maybe conclusion sections. t really needs a better ending to the discussion and a better conclusion to relate this more to 'food' and human benefits. Too much detail on the fish metabolism and not sufficient towards human health such as the omega 3 fatty acids elevation in the muscle. Please elaborate more.
Response 4:Thank you for your valuable feedback. I understand your suggestion to better link this study with consumers, retailers, and commercial advantages, as well as the need to strengthen the connection between the discussion and human health.In response to your comment, since this study primarily focuses on the impact of farming systems on fish metabolism, it has limited coverage of cost-benefit analysis and commercial models. Therefore, the paper does not delve deeply into cost analysis or the consumer perspective. However, in response to your suggestions, I have included a discussion of the limitations of this study, particularly regarding market evaluation and economic feasibility, and emphasized the need for future comprehensive assessments of these farming systems, including their long-term economic performance and market feasibility.For the conclusion section, I have also included a future outlook to ensure that the results highlight the practical commercial and health benefits of these farming systems for consumers, retailers, and industry professionals.
